# Concept Reachability in Diffusion Models: Beyond Dataset Constraints

**Marta Aparicio Rodriguez** [1]   **Xenia Miscouridou** [1][2]   **Anastasia Borovykh** [1]

## Abstract

Despite significant advances in quality and complexity of the generations in text-to-image models, *prompting* does not always lead to the desired outputs. Controlling model behaviour by directly *steering* intermediate model activations has emerged as a viable alternative allowing to *reach* concepts in latent space that may otherwise remain inaccessible by prompt. In this work, we introduce a set of experiments to deepen our understanding of concept reachability. We design a training data setup with three key obstacles: scarcity of concepts, underspecification of concepts in the captions, and data biases with tied concepts. Our results show: (i) concept reachability in latent space exhibits a distinct phase transition, with only a small number of samples being sufficient to enable reachability, (ii) *where* in the latent space the intervention is performed critically impacts reachability, showing that certain concepts are reachable only at certain stages of transformation, and (iii) while prompting ability rapidly diminishes with a decrease in quality of the dataset, concepts often remain reliably reachable through steering. Model providers can leverage this to bypass costly retraining and dataset curation and instead innovate with user-facing control mechanisms.

## 1. Introduction

The scaling of diffusion models (Sohl-Dickstein et al., 2015; Song & Ermon, 2019; Ho et al., 2020; Song et al., 2021) has significantly expanded their capacity to store and generate vast amounts of complex concepts. While prompts have become the de-facto manner to control the model output, there are numerous examples when simply prompting falls short

(see Figure 1). In addition to learning the visual and spatial components of concepts in images, text-to-image models must correctly associate the concept in an image with its corresponding semantic description in the caption (Huang et al., 2023; Ghosh et al., 2023; Chang et al., 2025). When this alignment fails, even overspecifying and re-prompting may fail to generate the target image.

Prior work has shown that, as an alternative to prompt-based sampling, one can operate directly on a model's representation level (Kwon et al., 2023; Ilharco et al., 2023; Wang et al., 2024; Wu et al., 2024). In particular, by editing specific activations, the sampling trajectory of diffusion models can be adjusted towards a particular target (Epstein et al., 2023; Samuel et al., 2024; Li et al., 2024a). While these works show that it *is* possible to steer towards certain output concepts, we lack a concrete framework to understand the complexity of guidance, or in other words the **reachability of concepts**. When can a concept be reliably accessed through prompting? If prompting fails, under what conditions can steering reliably reach the concept? And ultimately, what factors render a concept entirely unreachable despite it being in some way present in the training data?

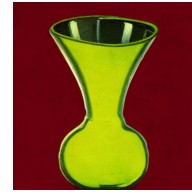 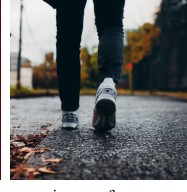 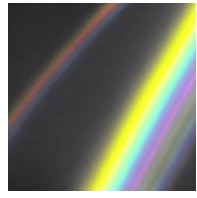

an image of a beaker    an image of a person walking left    an image of a rainbow in black and white

Figure 1: Images generated by Stable Diffusion (Rombach et al., 2022) that fail to produce the desired outcome due to hypothesised dataset limits: (L) a scarce concept, (C) underspecification in a caption, (R) biases. See Appendix F.2 for details.

Existing work has shown that the dataset structure plays a key role in reachability. For GANs and VAEs, Zhao et al. (2018) conclude that biases in the train set will influence the generation ability of models, regardless of the architecture or training algorithm implemented. In diffusion models, Chang et al. (2025) show that certain characteristics such as balanced datasets help mitigate the failure of a generated

[1]Department of Mathematics, Imperial College London, UK [2]Department of Mathematics and Statistics, University of Cyprus, Cyprus. Correspondence to: Marta Aparicio Rodriguez <marta.aparicio-rodriguez22@imperial.ac.uk>.

*Proceedings of the 42$^{nd}$ International Conference on Machine Learning*, Vancouver, Canada. PMLR 267, 2025. Copyright 2025 by the author(s).

image to match the target prompt. However, the complexity of the real world complicates the construction of a balanced training dataset that fully captures the true data-generating process. Figure 1 highlights scenarios in which prompting fails to produce the correct output. Underlying these scenarios are three core dataset limitations:

1. Scarcity of concepts: real-world datasets often exhibit an uneven distribution of concepts, with some being underrepresented or appearing infrequently. Such scarcity can hinder a model's ability to effectively learn these concepts.

2. Underspecification of captions: real-world datasets often include captions that do not describe *all* the concepts present in an image. This lack of detailed annotation can constrain a model's ability to accurately associate unmentioned concepts with their corresponding semantic representations.

3. Biases: real-world datasets frequently exhibit biases, where certain concepts consistently *co-occur*. This inherent correlation can make it challenging for models to disentangle these concepts, limiting their ability to independently generate or represent them.

Real-world images are often highly complex, with many intertwined factors that can make it challenging to isolate the exact conditions under which a model successfully reaches a concept. To address this, we work within a synthetic framework that allows us to systematically vary the structure of the dataset (for details, see Figure 2 and Section 3.3). This enables us to understand in detail how the mechanics of concept reachability are influenced by the above-mentioned three scenarios. To verify the generality of our main conclusions, we analyse the impact of the same scenarios on real-world data, including Stable Diffusion (Rombach et al., 2022) and CelebA (Liu et al., 2015). We list our contributions as:

- We show that even minor dataset limitations severely decrease the effectiveness of prompts, highlighting an inherent weakness in relying solely on this method of model control.

- We demonstrate that concepts remain accessible within the latent space, even under highly corrupted dataset conditions.

- We demonstrate a phase shift in reachability: a rapid increase in reachability can be observed as the number of images containing a concept is increased beyond a concept-agnostic, low threshold.

- We identify when concepts cannot be reached: when a concept is not specified in the captions of the train set,

models are unable to disentangle, and hence reach, the concepts effectively.

Our work demonstrates both the limits of prompting and the *resilience* of latent space interventions under three commonly present dataset limitations. Our work suggests that instead of curating new datasets or retraining models from scratch, model providers can enable users to reach concepts through novel control mechanisms. By shifting focus from data curation to user-driven model steering, providers can enhance model usability, robustness, and accessibility in ways that go beyond the limitations of prompting.[1]

## 2. Prior Work

**Control in LLMs**  Large language models (LLMs), having been trained on vast amounts of data, can be conceptualised as powerful information compressors, raising the challenge of how to effectively extract task-specific information. Various methods have been proposed to address this challenge without requiring fine-tuning of all model parameters. Some approaches focus on fine-tuning a smaller subset of weights (Hu et al., 2021; Zaken et al., 2022; Ilharco et al., 2023; Wu et al., 2024), while others enhance performance by optimising the input for specific tasks (Liu et al., 2022). Existing work additionally explores modifying a model's activations by introducing steering vectors at specific layers (Meng et al., 2022; Todd et al., 2024; Panickssery et al., 2024; Marks & Tegmark, 2024; Turner et al., 2024; Li et al., 2024b). In particular, recent advancements in Sparse Autoencoders demonstrate their potential in capturing interpretable latent representations, enabling fine-grained control and enhanced interpretability in language models (Cunningham et al., 2023; Templeton et al., 2024).

**Steering text-to-image models**  Similar to LLMs, previous works have successfully steered diffusion models to generate rare concepts, compose concepts and manipulate specific attributes in an image. Samuel et al. (2024) optimise the initial random seed to address generation failures and produce desired outputs. Other methods focus on modifying the U-net output (Wang et al., 2024; Gandikota et al., 2025) or adjusting activations of particular layers. Research has shown that editing the generation process can be achieved by adding vectors or fine-tuning the cross-attention layers that inject semantic information into the model (Hertz et al., 2022; Epstein et al., 2023; Kumari et al., 2023; Gandikota et al., 2024). Furthermore, semantically meaningful directions in the bottleneck layer of the U-net have been identified as a means to steer the generation effectively (Kwon et al., 2023; Li et al., 2024a; Haas et al., 2024).

---

[1]Code is available at `https://github.com/martaaparod/concept_reachability`.

**Data attribution methods** The study of how training data influences a model's output (Koh & Liang, 2017; Pruthi et al., 2020; Park et al., 2023; Hammoudeh & Lowd, 2024), particularly in real data, is closely intertwined with the broader question of how dataset constraints govern reachability. Data attribution techniques have been extended to image generation in diffusion models by analysing the sampling dynamics (Georgiev et al., 2023; Zheng et al., 2024), or by examining intermediate checkpoints (Xie et al., 2024) or hidden representations obtained during the training process (Brokman et al., 2025). While these approaches provide valuable insights, a significant practical challenge persists: rigorously evaluating the accuracy of influence estimation methods. To address this, Wang et al. (2023) introduces a methodology for constructing datasets explicitly influenced by known datapoints, thereby establishing a framework for empirical evaluation. In our work, we adopt a synthetic dataset, where the influence relationships are known by design.

**Analysis of reachability in generative models** Underlying the question of whether we can reach a certain concept lies the ability of the model to properly compose concepts seen during training. Research assessing this compositionality through prompting concludes that models can compose latent factors in novel ways if trained on sufficiently diverse data or for extended periods (Deschenaux et al., 2024; Okawa et al., 2023). Studies have also explored the learning dynamics that shape a model's generalisation ability. Mészáros et al. (2024) observe a simplicity bias in LLMs, where the learning process prioritises simpler tasks earlier in training. Additionally, Park et al. (2024) identify sudden transitions where diffusion models rapidly acquire the ability to generate specific concepts. In practice, failure cases do remain, with the work of Chang et al. (2025) analysing the impact of the underlying data distribution and lack of coverage of unique phenomena. Furthermore, steering vectors in LLMs have been found to sometimes produce unreliable or even counterproductive results (Tan et al., 2024). Building on these insights, our work investigates the factors contributing to the unreliability of reachability methods.

# 3. Background

## 3.1. Diffusion Models

Denoising diffusion probabilistic models (Ho et al., 2020) approximate the distribution $p_{data}(\mathbf{x})$ that gives rise to a collection of data points $\mathcal{X}$. During training, noise is added to images $\mathbf{x}_0$ from a train set $\mathcal{X}$ to give latents $\mathbf{x}_t = \sqrt{\bar{\alpha}_t}\mathbf{x}_0 + \sqrt{1-\bar{\alpha}_t}\boldsymbol{\epsilon}_t$, for $t \in [0, T]$ and appropriate constants $\bar{\alpha}_t$ dependent on a noise schedule. A U-net (Ronneberger et al., 2015) $\boldsymbol{\epsilon}_\theta$ is trained to match the added noise by minimising the loss function

$$\mathcal{L} = \mathbb{E}_{t \sim [1,T], \mathbf{x}_0, \boldsymbol{\epsilon}_t} \|\boldsymbol{\epsilon}_t - \boldsymbol{\epsilon}_\theta(\mathbf{x}_t, t)\|^2.$$

We implement text-to-image diffusion models $\boldsymbol{\epsilon}_\theta(\mathbf{x}_t, t, y)$, that additionally condition generation on a text prompt $y \in \mathcal{Y}$, passed through a text encoder $\mathcal{E}$ and inputted into the U-net through cross-attention layers (Vaswani et al., 2017; Rombach et al., 2022; Ramesh et al., 2022; Nichol et al., 2022; Epstein et al., 2023). The train set is comprised of image-caption pairs $(\mathbf{x}, y) \in \mathcal{X} \times \mathcal{Y}$. The sampling process involves the denoising of a latent $\mathbf{x}_T \sim \mathcal{N}(\mathbf{0}, \mathbf{I})$ conditioned on an input prompt $y'$. See Appendix A for details on our choice of architecture and hyperparameters.

## 3.2. Concepts

We introduce assumptions on the underlying structure of our data, following a similar approach to Zhao et al. (2018); Llera Montero et al. (2021); Okawa et al. (2023); Park et al. (2024); Wang et al. (2024). We assume that the images in the dataset are generated by a set of factors, such as object identity, colour, position or texture. Each factor can take certain values, which we denote as *concepts*. Formally, we say there exists a set of $n$ concept variables $\mathbf{F} = \{F_1, F_2, \ldots, F_n\}$, that define the image $\mathbf{x} \sim p_{data}(\mathbf{x})$. Each of the variables $F_i$ are sampled from their respective distributions $p(F_i)$. We denote the set of possible values the variables $F_i$ can take as $\mathcal{F}_i$, which can be discrete or continuous. The *space of concepts* is then defined as the set $\mathcal{F} = \mathcal{F}_1 \times \mathcal{F}_2 \times \cdots \times \mathcal{F}_n$ containing all possible combinations the factors can take.

Each combination of values $(f_1, f_2, \ldots, f_n) \in \mathcal{F}$ uniquely determines an image $\mathbf{x}$, where this relation is defined by an injective function $g : \mathcal{F} \to \mathbb{R}^{W \times H \times C}$ that transforms the tuple $(f_1, f_2, \ldots, f_n)$ into an image $\mathbf{x}$ with the target concepts $f_1, f_2, \ldots, f_n$. This function determines the distribution of $\mathbf{x}$ when conditioned on a combination of factors.

Throughout our work, we focus on a subset of factors assumed to be identifiable for all images sampled from $p_{data}(\mathbf{x})$. Without loss of generality, we assume these are the first $m < n$ factors $\mathcal{F}_1, \mathcal{F}_2, \ldots, \mathcal{F}_m$. We refer to the corresponding values of these factors, $(f_1, f_2, \ldots, f_m)$, as *concepts of interest*. We use the notation $[f_{i_1}, f_{i_2}, \ldots, f_{i_l}]_{\mathcal{X}}$, where $i_k \in \{1, 2, \ldots, m\}$ for $k = 1, 2, \ldots, l$, to refer to subsets of the dataset that share the concepts $f_{i_1}, f_{i_2}, \ldots, f_{i_l}$ for the factors $\mathcal{F}_{i_1}, \mathcal{F}_{i_2}, \ldots, \mathcal{F}_{i_l}$, and have no restrictions along other factors.

We further assume that captions capture the expressiveness of the concepts of interest in an image. For an image $\mathbf{x}$ containing concepts of interest $(f_1, f_2, \ldots, f_m)$, we assume there is an injective function $h : \mathcal{F}_1 \times \mathcal{F}_2 \times \cdots \times \mathcal{F}_m \to \mathbb{R}^l$ such that $h(f_1, f_2, \ldots, f_m)$ is a string containing the semantic information relevant to the concepts $f_1, f_2, \ldots, f_m$.

Therefore, image-caption pairs $(\mathbf{x}, y) \in \mathcal{X} \times \mathcal{Y}$ satisfy that the concepts of interest of $\mathbf{x}$ and the tuple $h^{-1}(y)$ are equal.

## 3.3. Problem Setting

Our study examines how the presence and relationships within training data influence the ability of diffusion models to reach concepts (Figure 2). To achieve this, we systematically vary the dataset used to train a model, and observe the evolution of reachability for specific concept combinations. We vary the following:

1. Scarcity of concepts: starting with a balanced dataset, the presence of individual concepts is progressively reduced. We evaluate how the reachability of combinations containing the underrepresented concept is affected.

2. Underspecification of captions: beginning with a fully annotated dataset, semantic information relevant to certain factors is removed. The impact of this specification reduction on the model's ability to reach complete concept combinations is assessed.

3. Biases: from a dataset where two concepts are consistently paired, we incrementally introduce data containing only one of the concepts. We analyse how the model's ability to independently reach each concept evolves.

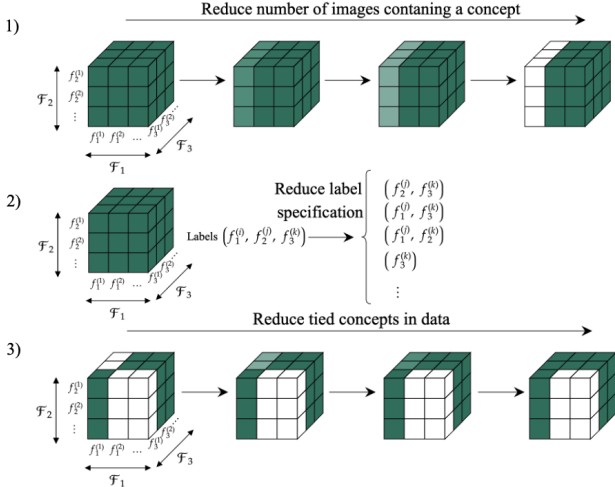

Figure 2: Visualisation of the structure of the dataset according to concepts of interest (in the diagram, three). Each block in the cube represents the collections of images in $[f_1^{(i)}, f_2^{(j)}, f_3^{(k)}]_{\mathcal{X}}$. A darker shade represents a higher number of images in the block. Data modifications are as described in Section 3.3.

## 3.4. Reachability

In this section, we introduce the notion of reachability and key definitions used in subsequent sections.

**Definition 3.1** (Concept Reachability). Given a target combination of concepts $(f_1, f_2, \ldots, f_m)$ and method $M$ to access these, we define the *reachability* of $(f_1, f_2, \ldots, f_m)$ as the *accuracy* or proportion of images produced by $M$ that contain the concept combination $(f_1, f_2, \ldots, f_m)$.

We also identify cases during generation in which outputs are out-of-distribution.

**Definition 3.2** (Out-of-distribution). Given a model trained on a dataset $\mathcal{X} \times \mathcal{Y}$ with concept function $g$ as defined in Section 3.2, a generated image $\mathbf{x}^*$ with concepts of interest $(f_1^*, f_2^*, \ldots, f_m^*)$ is out-of-distribution (OOD) if $[f_1^*, f_2^*, \ldots, f_m^*]_{\mathcal{X}} = \emptyset$.

We additionally distinguish between two mechanisms that result in unseen combinations. This distinction provides a valuable framework for understanding differences in reachability levels in OOD generalisation scenarios, as in Sections 5.2 and 5.4. In particular, we consider a model's ability to combine known factors into a new configuration and model's ability to transfer knowledge of attributes from one positional context to another.

**Definition 3.3** (Compositionally out-of-distribution). Given an out-of-distribution combination of concepts $F_o = (f_1, f_2, \ldots, f_m) \in \mathcal{F}_1 \times \mathcal{F}_2 \times \cdots \times \mathcal{F}_m$, we say it is *compositionally out-of-distribution* if for every concept $f_j$ in the combination, there exists a concept combination in the train set whose $j$th component is $f_j$.

**Definition 3.4** (Positionally out-of-distribution). Given an out-of-distribution combination of concepts $F_o = (f_1, f_2, \ldots, f_m) \in \mathcal{F}_1 \times \mathcal{F}_2 \times \cdots \times \mathcal{F}_m$, we say it is *positionally out-of-distribution* if it is not compositionally out-of-distribution, and there exists a permutation $\rho : \{1, 2, \ldots, m\} \to \{1, 2, \ldots, m\}$ such that $F_o' = (f_{\rho(1)}, f_{\rho(2)}, \ldots, f_{\rho(m)})$ is seen during training.

# 4. Methodology

## 4.1. Dataset

We design a synthetic experimental setup that pursues controllability over the tasks outlined in Section 3.3. Our setup is similar to the work of Zhao et al. (2018); Llera Montero et al. (2021); Scimeca et al. (2022); Wiedemer et al. (2023); Okawa et al. (2023); Deschenaux et al. (2024); Park et al. (2024); Chang et al. (2025). We use a dataset that contains images of coloured shapes on a black background, where one shape is partially covered by the other (Figure 10). See Appendix B for further details.

Let $\mathcal{S} = \{$circle, triangle, square$\}$ and $\mathcal{C} = \{$red, green

blue}. The images in our dataset are labelled using the captions, "a $\{c_1\}$ $\{s_1\}$ behind a $\{c_2\}$ $\{s_2\}$", where $s_1, s_2 \in \mathcal{S}$, and $(c_1, c_2) \in (\mathcal{C} \times \mathcal{C}) \setminus \{(c, c) : c \in \mathcal{C}\}$. That is, all combinations of any two shapes and any two colours in $\mathcal{S}$ and $\mathcal{C}$ are admitted, excluding images containing two shapes of the same colour. We refer to the concepts of interest of images as $(c_1, s_1, c_2, s_2)$ (Figure 3). We create the dataset such that each of the combinations of concepts $(c_1, s_1, c_2, s_2)$ is originally seen an equal number of times.

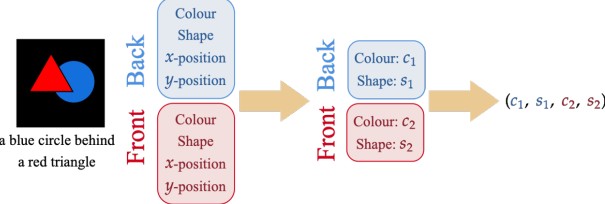

Figure 3: Concepts of interest in the dataset. The concepts of an image are summarised as the tuple $(c_1, s_1, c_2, s_2)$, where each of these positions refers to back colour, back shape, front colour and front shape respectively. In the diagram, this is $(blue, circle, red, triangle)$.

Our original dataset is comprised of 54 combinations of shapes and colours $(c_1, s_1, c_2, s_2)$, each containing 1000 images. During experiments where the number of images containing certain concept combinations is varied, we preserve the total size of the dataset by adjusting the size of the remaining combinations.

## 4.2. Steering

We employ implementations that add a constant vector to a specific layer of the U-net during sampling. These methods consist of two stages: the optimisation of a concept vector using images that contain the target concept combinations, and the addition of the optimised concept vector during sampling. Motivating our choices from existing work on steering (Kwon et al., 2023; Kumari et al., 2023; Li et al., 2024a), we consider two spaces in which to implement steering: the text encoding of the prompt and the bottleneck layer of the U-net.

Let $\boldsymbol{\epsilon}_\theta$ denote a conditional U-net trained on the dataset $\mathcal{X} \times \mathcal{Y}$. For a given combination of concepts $(f_1, f_2, \ldots, f_m)$, the model can generate an image containing this combination by using the prompt $y_e = h(f_1, f_2, \ldots, f_m)$. Alternatively, we sample from a starting prompt $y_s$ and steer the generation process towards the desired combination of concepts $(f_1, f_2, \ldots, f_m)$. Note that $y_s$ may not satisfy $y_s = y_e$, or that the image produced by the model may not accurately represent the target concepts of interest. Thus, we use steering to enable the model to reach the desired output more accurately. Let $\mathcal{Z}$ be a collection of images

containing the concept combination of interest. Using the starting prompt $y_s$, we create the image-label pairs $(\mathbf{x}_0, y_s)$ for every $\mathbf{x}_0 \in \mathcal{Z}$.

**Prompt space** Let $\mathcal{E}(y_s)$ be the text embedding outputted by the text encoder for the prompt $y_s$. We modify the sampling trajectory of $y_s$ by replacing $\mathcal{E}(y_s)$ by $\mathcal{E}(y_s) + \mathbf{v}_p$, where $\mathbf{v}_p$ is a vector of the same dimensionality as $\mathcal{E}(y_s)$. We refer to the output of the U-net after this modification as $\boldsymbol{\epsilon}_\theta(\mathbf{x}_t, t, y, \mathbf{v}_p)$ (Figure 9A). To obtain the vector $\mathbf{v}_p$, the following loss is minimised:

$$L_p = \mathbb{E}_{t \sim [1,T], (\mathbf{x}_0, y_s), \boldsymbol{\epsilon}_t} \| \boldsymbol{\epsilon}_t - \boldsymbol{\epsilon}_\theta(\mathbf{x}_t, t, y_s, \mathbf{v}_p) \|^2.$$

Note that the weights of $\boldsymbol{\epsilon}_\theta$ are frozen. The vector $\mathbf{v}_p$ therefore accounts for the mismatch between the semantic information contained in $y_s$ and the visual information in the noisy latents obtained from images in $\mathcal{Z}$.

**h-space** We also consider steering of the starting prompt $y_s$ on the $h$-space, following the implementation of Li et al. (2024a). Let $\boldsymbol{\epsilon}_\theta(\mathbf{x}_t, t, y_s, \mathbf{v}_h)$ denote the output of the U-net when inputted the noisy image $\mathbf{x}_t$, the prompt $y_s$ and the vector $\mathbf{v}_h$, which is added to the bottleneck layer output (Figure 9B). The vector $\mathbf{v}_h$ is optimised to minimise the loss:

$$L_h = \mathbb{E}_{t \sim [1,T], (\mathbf{x}_0, y_s), \boldsymbol{\epsilon}_t} \| \boldsymbol{\epsilon}_t - \boldsymbol{\epsilon}_\theta(\mathbf{x}_t, t, y_s, \mathbf{v}_h) \|^2.$$

Similarly to steering on the prompt space, the vector $\mathbf{v}_h$ is optimised to reconcile the mismatch between the semantic information in $y_s$ and the visual concepts in the images.

## 4.3. Evaluation Method

Reachability, as defined in Section 5, necessitates the identification of the concepts of interest within sampled images. Identifying concepts in images on a large scale can prove a challenging task, with most real-data methods relying on vision-text models such as CLIP (Radford et al., 2021; Huang et al., 2023; Ghosh et al., 2023). To evaluate the concepts within the generated images, we train three classifiers for identifying the back shape, front shape and back-front colour pairs of the two shapes in each image. The resulting labels are then compared against the target concepts. Images producing incomplete outcomes (such as a black background with no shapes or an image only showing one shape) are accounted for as incorrect images (see Appendix C). Throughout our experiments, we train four models initialised with different random seeds and report the mean results obtained.

# 5. Empirical Results

## 5.1. Baseline: Balanced Dataset

Before introducing additional complexities to the data, we establish a clear understanding of the different reachability methods under balanced conditions. Specifically, we train diffusion models and use steering to generate various target combinations, varying the level of complexity between these and the starting prompt. Reachability when directly prompting the target combination is also accounted for, as illustrated in Figure 4.

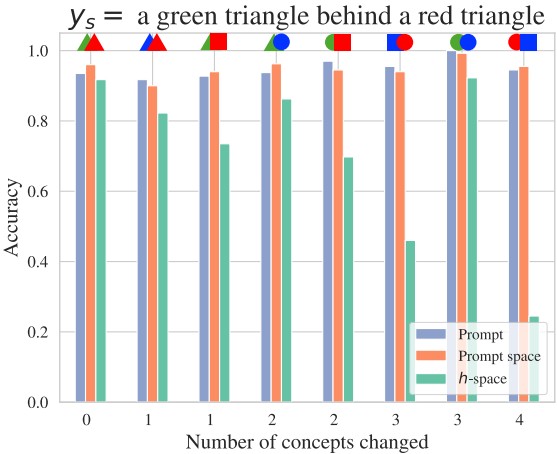

Figure 4: Reachability to different concept combinations when prompting, as well as steering from the starting prompt $y_s$ ="a green triangle behind a red triangle". Target colour-shape combinations (note that the relative position is representative) are shown at the top of each bar, and are organised according to the number of concepts that differ from the concepts of $y_s$.

**Prompt-space steering matches prompting reachability** Prompting and steering on the prompt space yield an accuracy above 0.9, demonstrating their effectiveness in reaching concepts under balanced conditions. This highlights steering as a reliable method for accessing desired concepts. In contrast, steering within the $h$-space exhibits a pronounced dependency on the targeted concepts (for fixed $y_s$). Notably, accuracy on the $h$-space diminishes as the number of modified concepts increases, suggesting a limitation in its capacity to handle heavily complex modifications.

**Reachability bias when steering on the $h$-space** We observe a tendency for reachability to be higher, relative to other concept combinations with the same number of modified concepts, when the target front shape and back shape are identical (second column from the right of Figure 4). We identify this as a property of the dataset, as it is consistently observed across different models, and find it aligns with

the steerability bias reported by Tan et al. (2024). This supports the role of dataset properties in shaping reachability behaviour. We provide further analysis of reachability on the $h$-space in Appendix E.1.

Overall, we observe that prompting and steering perform effectively in a perfectly balanced dataset. In the following sections, we present results regarding their performance under adverse data conditions. Based on the results observed under balanced conditions, we choose the starting prompt that maximises performance when steering on the $h$-space. Unless stated otherwise, throughout the remaining experiments we choose to implement steering from the starting prompts $y_s$ which describe the target concepts (0 concepts changed).

## 5.2. Scarcity of Concepts

We fix $(red, triangle, green, square)$ as a target concept combination and gradually reduce the number of data points in the subset $[c_1 = red]_{\mathcal{X}}$. This process is repeated for the subsets $[s_1 = triangle]_{\mathcal{X}}$, $[c_2 = green]_{\mathcal{X}}$, and $[s_2 = square]_{\mathcal{X}}$. We present two of these tests in this section and relay the remaining cases to Appendix E.3.

**Reachability drops sharply past a critical threshold** We implement steering to our chosen target concept combination and observe that, although reachability decreases as the presence of concepts is reduced, this decline remains gradual until reaching a threshold at $p_{\mathcal{X}}(f) \approx 0.01$, $f = c_1, s_2$. Below this, reachability decreases rapidly (Figure 5). This decline is less pronounced for steering, most noticeably for $s_2 = square$, highlighting that *steering can be more effective at maintaining concept reachability under conditions of scarcity*. This behaviour suggests a phase transition-like effect: when concept presence falls below 1%, the system shifts abruptly from a state of high reachability to one of significantly diminished reachability. This phase transition implies that the model requires few data points, relative to the train set size, in order to learn a concept effectively. Increasing the number of data points containing a concept beyond this critical threshold has limited impact on improving reachability.

**Positionally OOD combinations can be reachable through steering** When the presence of a concept $f$ is such that $p_{\mathcal{X}}(f) = 0$, generating the combination $(red, triangle, green, square)$ results in a positionally OOD target. For instance, eliminating the data points in $[c_1 = red]_{\mathcal{X}}$, limits the model's exposure to red shapes in the back position, even if red shapes remain present in the front position $c_2$. Our findings reveal that models are unable to reach the target concept combination through prompting (Figure 5), aligning with observations by Chang et al. (2025). In contrast, steering methods can achieve moder-

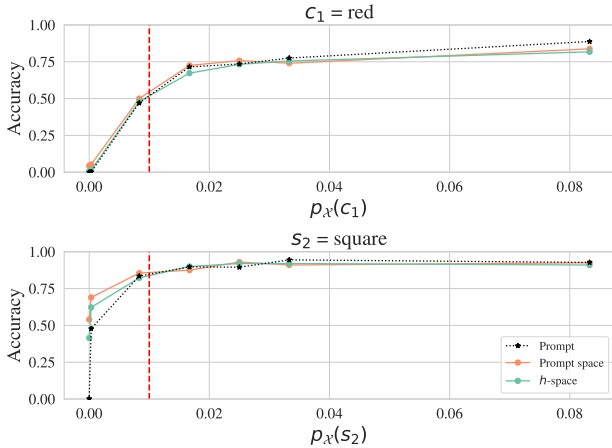

Figure 5: Accuracy of prompting and steering on the prompt space and $h$-space to $(red, triangle, green, square)$ for starting prompt $y_s =$ "a red triangle behind a green square" and varying proportion $p_{\mathcal{X}}$ of images in the dataset containing the concepts $c_1 = red$ (top) and $s_2 = square$ (bottom) across the dataset. The dashed red line marks the approximate threshold 0.01 of the shift in reachability.

ate reachability. Notably, when reducing $[s_2 = square]_{\mathcal{X}}$, steering on the prompt space achieves more than 50% accuracy, even when no training images contain the reduced concept. This indicates that when prompting fails, alternative mechanisms may enable models to generalise concepts to unseen positions.

## 5.3. Underspecification of Captions

We reduce the factors of interest of the dataset, initially $(c_1, s_1, c_2, s_2)$. For instance, removing the specification of $c_1$ results in captions $y$ containing information only with respect to $(s_1, c_2, s_2)$. See Appendix D for details. We then evaluate the reachability of $(c_1, s_1, c_2, s_2)$ via prompting and steering.

We present the results of prompting when describing the full set of target concepts. For example, "a red triangle behind a green square". Steering is performed on top of starting prompts $y_s$ that reflect only the concepts seen during training. For example, when removing back colour $(c_1)$, steering is implemented from $y_s =$ "a triangle behind a green square" to images with the concept combination $(red, triangle, green, square)$.

**Steering outperforms prompting under low specification**  In all cases of specification reduction, we observe an improvement in reachability with respect to prompting when using either steering method (Figure 6a). Among these, steering on the prompt space achieves the highest reachability, demonstrating it is a more effective approach

for accessing target concepts under reduced specification conditions.

**A decrease in specification hinders reachability**  Despite the improvement in reachability achieved through steering, reducing the specification of captions during training results in a rapid decrease in reachability. Figure 6a shows average accuracy falls below 0.30 for models trained with fewer than two concepts specified. Classifying the outputs after prompting and steering reveals that models tend to generate concepts accurately for the factors specified during training, but fail to correctly generate the missing concepts. An example for one random seed is presented in Figure 6b. When prompting is used, the model's accuracy on specified labels is high and on unspecified labels is nearly equivalent to chance-level performance. In contrast, steering yields higher accuracy on the full target concept combination, although remaining close to the prompting baseline. These findings show captions hold a pivotal role in organising and structuring a model's latent space, providing a semantic framework that enhances its ability to access concepts. Our results demonstrate that more detailed labeling of training images can improve the success rate of generating target concept combinations. On the contrary, given a trained model, overly detailed prompts may not lead to an improvement of reachability.

We further analyse reachability when using $y_s$ containing the full semantic information of the target combination $(c_1, s_1, c_2, s_2)$ in Appendix E.4.

## 5.4. Biases

We measure reachability to individual concepts that are consistently paired during training. Specifically, we construct a dataset where an image in the training set satisfies $c_1 = blue$ if and only if $s_1 = circle$. To evaluate the model's ability to disentangle these concepts, we measure reachability to concept combinations of the form $(blue, s'_1, c_2, s_2)$, where $s'_1 \neq circle$, and $(c'_1, circle, c_2, s_2)$, where $c'_1 \neq blue$. Prompt space steering exhibited minimal variation across different starting prompts $y_s$, whereas on the $h$-space, starting with a prompt describing the target concepts of interest yielded the highest accuracy. We present this in Figure 7.

**Biases in compositional OOD generalisation can be bypassed through steering**  When no train images contain only blue in the back, attempting to reach combinations where either concept appears separately places the model in a compositionally OOD setting. Figure 7 shows steering, particularly on the prompt space, achieves a moderately high reachability to both concepts (accuracy of black X's), suggesting that models are capable, to some extent, of disentangling heavily biased concepts. Prompting fails to achieve this level of disentanglement, revealing its inability to accu-

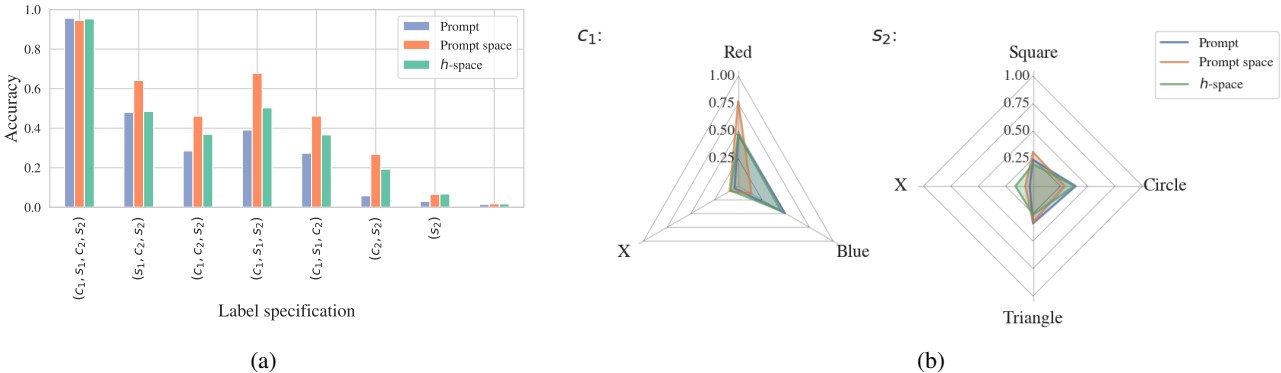

(a)                                                                                    (b)

Figure 6: a) Average reachability across 10 randomly chosen concept combinations $(c_1, s_1, c_2, s_2)$ for different levels of label specification. b) Outputs produced by a model when steering towards $(red, circle, green, square)$ when the factors $c_1$ (left) and $s_2$ (right) are removed from the captions of the train set. Each diagram shows the target concept in the top axis, and the alternative concepts the removed factor can take in the remaining directions. The label $X$ represents any other output, including those that do not produce the target concept for the seen factors. Each diagram displays the proportion of images of the output belonging to each label. A reachability method that correctly generates the target concept combination will produce a high proportion of images on the top axis. If a model only fails to generate the removed concept value, the proportion of images at $X$ will be low.

rately reflect a model's full capabilities.

**Increasing the presence of an individual concept improves the separate reachability of both tied concepts** As the number of images containing non-circular blue shapes increases, the reachability to combinations only containing $c_1 = blue$ rises rapidly, evidenced by the large increase in accuracy along the horizontal axis in Figure 7. This shows a similar threshold pattern to that observed in Section 5.2. Interestingly, the lighter X's on the vertical axis tend to achieve the highest accuracy, suggesting that the reachability of non-blue circles in the back also increases, albeit with minor variability. This pattern implies that the model learns to identify the contrasting concept, likely by inferring its definition through what it is not.

See Appendix E.5 for additional results.

### 5.5. Reachability in Real Settings

We additionally implement steering on the prompt space of Stable Diffusion in order to explore how reachability is impacted in a real-world setting. Figure 8 presents samples from the same initial random seed $x_T$ as in Figure 1, with additional steering implemented. The resulting images show a clear improvement, more accurately representing the target concepts. Further examples and analysis of reachability through steering are provided in Appendix F.3. Overall, we observe that steering can extend beyond the reachability of prompting. However, dataset limitations such as underspecification continue to hinder reachability, exhibiting patterns consistent with those observed in our synthetic analysis.

Furthermore, we conduct a similar analysis to that in Section 5 using a subset of the CelebA dataset. Results are presented in Appendix G. Specifically, we use the labelled attributes for Gender (male/female) and Hat (wearing/not wearing a hat) as the concepts to be analysed. Overall, we again find that the structure of the training data strongly influences concept reachability. In particular, reachability declines sharply when the number of training examples containing a given concept falls below a certain threshold, and underspecified labels substantially hinder access to those concepts. Additionally, we observe that reducing biases in the co-occurrence of concepts improves a model's ability to represent and generate individual concepts in isolation. These findings are consistent with those observed in the synthetic setting, supporting the generality of these conclusions and highlighting their relevance in real settings.

## 6. Discussion and Conclusion

This paper explores the limits of reachability in diffusion models, examining the challenges in accessing target concepts. While prompting is the default approach, our findings reveal that it often fails to fully capture a model's potential for generating specific concepts. We demonstrate that steering can enhance the reachability of scarce concepts and disentangle biases inherent in datasets. We also identify key patterns in reachability such as shifts in a model's ability to access concepts and the critical role of proper concept specification in captions. This work provides a valuable framework for evaluating when concepts are reachable, and offers insights into how we can design train sets to improve

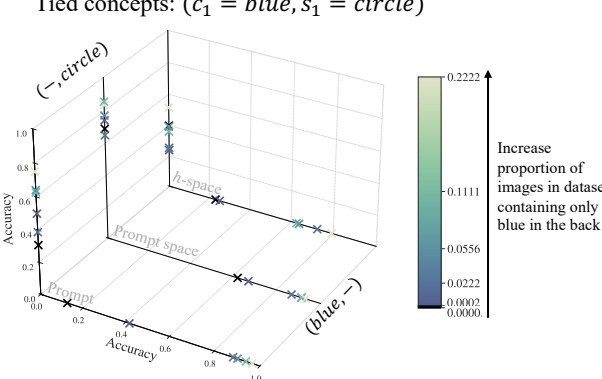

Tied concepts: ($c_1 = blue, s_1 = circle$)

Figure 7: Reachability to concept combinations containing non-blue circles in the back (vertical axis) and blue non-circular shapes in the back (horizontal axis). Results are averaged over 6 randomly chosen target concept combinations, with starting prompt $y_s$ describing those concepts. The X's mark the accuracy of reaching either type of concept when prompting, steering on the prompt space and steering on the $h$-space. A lighter colour represents a stronger presence of images containing only blue in the back in the dataset.

the generative performance of diffusion models.

The improvement observed in reachability through steering suggests a misalignment between the model's semantic and visual understanding of concepts. That is, a model may be able to reach a concept, but the prompt fails to access the correct spatial information required to generate it. Moreover, we observe differences in the accuracy of steering on the prompt space and steering on the $h$-space, which highlight the importance of the stage of the transformation at which steering is performed. In particular, the nature of the space in which steering is implemented will impact reachability. For example, while the text encoding is independent of the timestep $t$, the bottleneck output ($h$-space) depends on

it. The two spaces also differ in dimensionality, and the degree of disentanglement of either space may also influence steering effectiveness. We note that, while steering on the $h$-space does not perform optimally for every choice of $y_s$, our results show it is an effective method when $y_s$ aligns with the target concept combination.

An intriguing aspect of our experiments concerns a model's ability to employ different mechanisms (compositional or positional) to achieve OOD generalisation. As presented in Section 5.2, models face significant challenges in OOD positional generalisation, with instances where reachability across all methods approaches zero. While biased compositional OOD generalisation remains a demanding task, we observe moderate levels of reachability using at least one of the methods studied (Section 5.4). Generalising shape positionally from back to front is particularly interesting, as it requires the model to synthesise information from multiple images to reconstruct a complete, unobstructed shape. We detect no substantial variance in reachability when positionally generalising in either direction, suggesting that models find both tasks equally demanding.

Despite improvements in reachability, the steering methods explored in this study rely on an auxiliary collection of images $\mathcal{Z}$ containing the target concept combinations, which in real data may be hard to acquire. Investigating alternative steering methods that mitigate this dependency and enhance reachability remains an important direction for future research. Additionally, there may exist simple or more efficient alternative methods to steering that outperform existing approaches and warrant further exploration. Moreover, while our work isolates the impact of each obstacle studied, in real data it is likely that problems occur simultaneously and with varying degrees of complexity. Expanding our dataset to incorporate additional factors and more intricate relationships can provide a more comprehensive analysis of model behaviour.

## Acknowledgements

MAR is supported by the Department of Mathematics, Imperial College London, through the Roth Scholarship.

## Impact Statement

This paper focuses on studying the robustness methods for model control. It holds direct value for model providers as it highlights the delicacy of prompting and instead motivates the usage of alternative latent control methods.

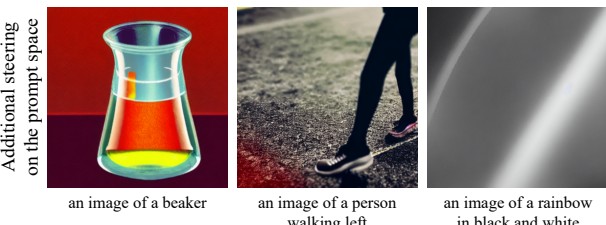

an image of a beaker    an image of a person walking left    an image of a rainbow in black and white

Figure 8: Images sampled from the same random seeds used in Figure 1 with steering vectors added to the prompt space during sampling. Using steering, we observe: (L) a more accurate representation of a beaker, (C) a successful change in orientation of the person towards the left, (R) the rainbow arc generated in greyscale tones.

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

# A. Hyperparameters

### A.1. Architecture

For our experiments, we used the Diffusers package (von Platen et al., 2022) to implement the U-net architecture. Our model comprises 3.7 million parameters and features a symmetric architecture with four downsampling and upsampling blocks. The channels in each block are set to $(16, 32, 64, 128)$, respectively. Each block contains two ResNet blocks.

Additionally, we incorporate cross-attention layers in the midblock, enabling the model to integrate information from a text-encoding input. For processing the text prompt, we use a pre-trained T5Small text encoder (Raffel et al., 2020). The encoder's weights are frozen throughout training of the U-net and during the optimisation of concepts vectors.

### A.2. Training

Training of the U-net is performed for 70 epochs using Adam with learning rate $0.001$ and default parameter values. Additionally, we use an exponential learning rate scheduler with parameter $gamma = 0.98$. All models are trained using $T = 1000$ and sampled with a DDPMScheduler at inference time.

### A.3. Concept Vector Optimisation

Concept vectors are initialised at the zero-vector, and optimised for 5000 steps using Adam, with learning rate 0.02 and default parameter values. This choice is observed to be sufficient for the optimised losses $L_p$ and $L_c$ to converge. A diagram of the different spaces in which the optimisation is implemented is shown in Figure 9.

For steering on the prompt-space, $\mathbf{v}_p$ is of the same dimensionality as the input prompt $y_s$. In the original dataset with captions "a $\{c_1\}$ $\{s_1\}$ behind a $\{c_2\}$ $\{s_2\}$", this accounts for $\mathbf{v}_p \in \mathbb{R}^{1 \times 10 \times 512}$, and is adjusted for the experiments in Section 5.3.

The dimension of the vectors $\mathbf{v}_h$ optimised in the bottleneck of the U-net is $\mathbf{v}_h \in \mathbb{R}^{128 \times 8 \times 8}$ for all experiments.

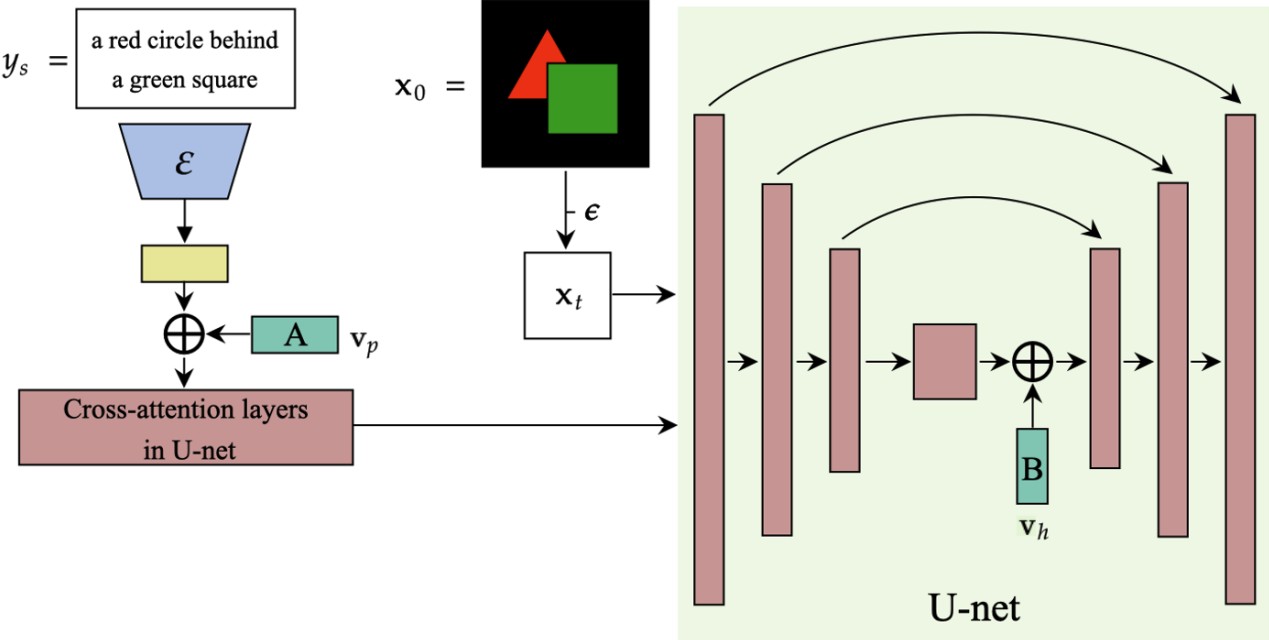

Figure 9: Diagram of spaces in the architecture where steering is implemented. A: prompt space, the concept vector is added to the encoding for the text prompt before passing through the cross-attention layers of the U-net. B: $h$-space, the concept vector is added to the bottleneck layer of the U-net, after the mid-block.

## B. Dataset Creation

Images are of dimension $64 \times 64$ pixels, comprised of two coloured shapes. To create the complete dataset, we generate 1000 images with each of the combinations of the concepts of interest $(c_1, s_1, c_2, s_2)$. Each image is created using the Pillow package in Python (Clark, 2015). First, the location of the centre of the back shape is determined, and then the position of the front shape is sampled uniformly from a neighbourhood of this centre. This neighbourhood ensured a reasonable portion of the back shape was always visible for every pair of shapes (thus, facilitating the classification task). The minimum percentage of each back shape visible in the balanced train set for the diffusion models is the following: circle 52.46%, triangle 48.97%, square 59.38%. Examples of train images are presented in Figure 10.

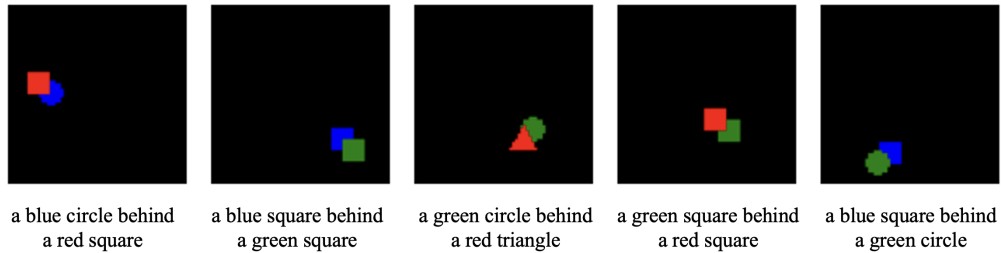

| a blue circle behind a red square | a blue square behind a green square | a green circle behind a red triangle | a green square behind a red square | a blue square behind a green circle |

Figure 10: Examples of image-caption pairs from our synthetic dataset. Each image shows two shapes of different colours, one partially covered by the other, on a black background.

When reducing the size of some of the combinations of the dataset, the size of the remaining combinations is proportionally increased to ensure that the total dataset size is as close as possible to 54,000 while maintaining uniformity (equal number of images) among the unaffected concept combinations.

## C. Classifier and Evaluation Details

We train three different classifiers for identifying the back shape, front shape and back-front colour pairs. The architecture consists of two convolutional layers that increase channel size to 16 and 32 respectively. Both consist of $3 \times 3$ kernels with ReLU activation and $2 \times 2$ max-pooling. The flattened output is passed to a fully connected layer of 128 units and ReLU. The output vector is three-dimensional in the case of both shape classifiers, and consists of six dimensions for the classifier of back-front colour pairs.

Each classifier is trained using Adam with learning rate $0.001$ on a dataset comprised of the original 54,000 balanced dataset and an equal number of sampled images from trained diffusion models. Training is implemented for 7 epochs. When evaluating on 5400 human-labelled images that were generated by prompting from two additional diffusion models trained on balanced data, the classifier obtains an accuracy of 96.63%.

During evaluation of reachability, 100 images are sampled using a reachability method. Each of these is labelled according to the outputs of the trained classifiers. Images are accounted for as correct if the labels match the target concept combination. In cases where classifier results were uncertain or ambiguous, additional human evaluations were conducted. Furthermore, we identify the number of (non-black) colours in an image, and if this number if distinct from 2, we account for the images as incorrect. After this, the proportion of correct images is used as the reachability value.

## D. Reduction of Label Specification on Captions

Our dataset is originally comprised of captions of the form "a $\{c_1\}$ $\{s_1\}$ behind a $\{c_2\}$ $\{s_2\}$". In order to reduce the semantic information of the prompt, we implement the following:

1. Remove $c_1$: replace the caption with "a $\{s_1\}$ behind a $\{c_2\}$ $\{s_2\}$"

2. Remove $s_1$: replace the caption with "a $\{c_1\}$ shape behind a $\{c_2\}$ $\{s_2\}$"

3. Remove $c_2$: replace the caption with "a $\{c_1\}$ $\{s_2\}$ behind a $\{s_2\}$"

4. Remove $s_2$: replace the caption with "a $\{c_1\}$ $\{s_2\}$ behind a $\{c_2\}$ shape"

5. Remove $c_1$ and $s_1$: replace the caption with "a $\{c_2\}$ $\{s_2\}$"

6. Remove $c_1$, $s_1$ and $c_2$: replace the caption with "a $\{s_2\}$"

7. Remove $c_1$, $s_1$, $c_2$ and $s_2$: replace the caption with the empty string, ""

# E. Additional Experiments

### E.1. Additional Analysis on a Balanced Data

**Final norm of optimised concept vector is indicative of reachability when steering on the $h$-space**   The differences in the accuracies achieved through steering on the $h$-space suggest that, from a given starting prompt $y_s$, certain concept combinations are inherently more reachable than others. While the final loss at the end of the optimisation of the steering vector might be considered a natural indicator of reachability via steering (lower loss associated with higher reachability), we observe cases where high final loss values correspond to high steering accuracies (Figure 11, teal x's). Our findings indicate that the final vector norm provides a more reliable indicator, with a larger norm being associated with lower reachability (Figure 11, bottom right). Interestingly, when sampling, adding a larger concept vector norm indicates a greater deviation from the latents that would be obtained from solely prompting $y_s$. Hence, we conclude that reachable concepts remain close in the $h$-space. Steering on the prompt space does not exhibit such patterns (Figure 11, left).

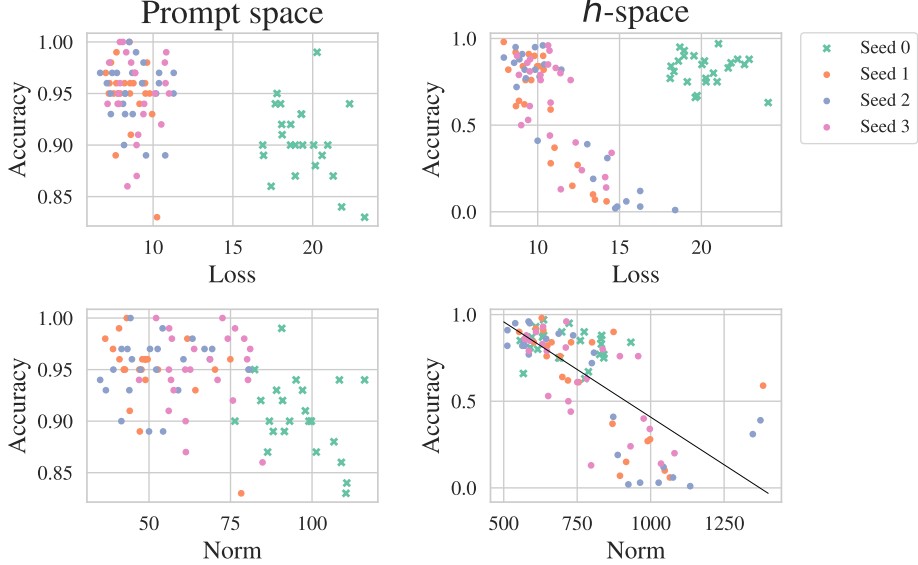

Figure 11: Relation between accuracy of steering on the prompt space and $h$-space and final loss (top row) and final vector norm (bottom row) after the optimisation of the concept vector. The different colours show the results for four different models trained on different random seeds.

### E.2. Reachability When Removing One Concept Combination from the Dataset

We test the ability of a model to generalise compositionally OOD in the simplest scenario, where only the target concept combination is removed from the dataset. Three examples of this are presented in Figure 12. Our results highlight the variability of accuracy of steering on the $h$-space, also observed in Section 5.1. Moreover, we observe that reachability is generally high and unaffected from the removal of the target combination. Prompting achieves the highest reachability, with steering on the prompt space also providing similar accuracy values. Thus, models are capable of generalising OOD for simple compositional concept combinations based on their ability to recombine learned individual concepts. We note that this is not necessarily extensible to any compositional OOD generalisation, as our experiments in Section 5.4 provide a

more complex compositional OOD generalisation task, where models are affected by the biases of combinations seen during training.

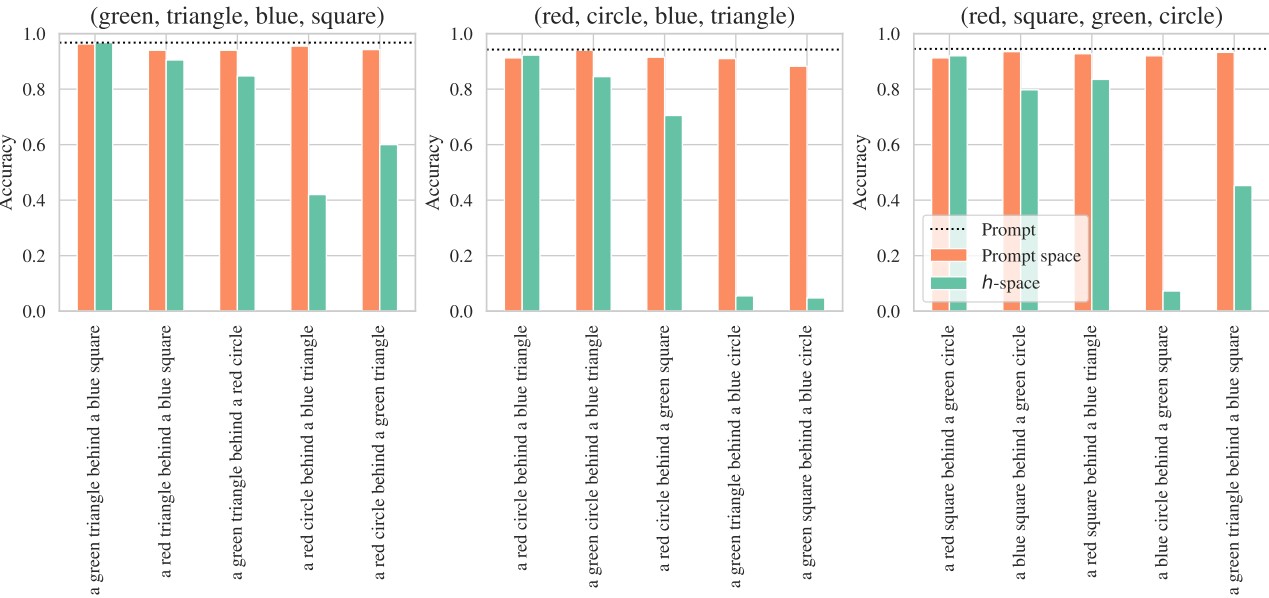

Figure 12: Steering accuracies for different starting prompts $y_s$ to $(green, triangle, blue, square)$, $(red, circle, blue, triangle)$ and $(red, square, green, circle)$ respectively. For each graph, the target concept combination is the only concept combination not present in the training dataset.

### E.3. Additional Results for Scarcity of Concepts

In this section we present additional results for Section 5.2. Figure 13 shows reachability results for different starting prompts $y_s$ when reducing $[c_1 = red]_{\mathcal{X}}$ and $[s_2 = square]_{\mathcal{X}}$. As observed in the baseline analysis, steering on the prompt space presents a more stable outcome, while steering on the $h$-space shows greater variability. The highest reachability values on the $h$-space are achieved on the starting prompt $y_s =$ "a red triangle behind a green square", that mentions the same concepts as those in the target combination. Note that despite the prompt $y_s =$ "a red triangle behind a green circle" achieving comparable reachability in the case of reduction of $c_1$, this is not consistent in the reduction of the presence of other concepts.

We also present the effect of decreasing the size of the subsets of the dataset $\mathcal{X}$: $[s_1 = triangle]_{\mathcal{X}}$, $[c_2 = green]_{\mathcal{X}}$. Figure 14 shows a similar sudden decrease in reachability, as observed Figure 5. Moreover, we highlight the improvement of reachability by steering on the prompt space over prompting below the threshold, most noticeably when reducing the concept $s_1 = triangle$.

### E.4. Additional Results for Underspecification

In this section we present additional results for analysing the effect of varying the number of label specification on reachability (Section 5.3). We implement steering from the same 10 randomly chosen target concept combinations and trained models as in Figure 6a, but with a starting prompt containing the full semantic information "a $\{c_1\}$ $\{s_1\}$ behind a $\{c_2\}$ $\{s_2\}$". This approach ensures that the size of the vector $\mathbf{v}_p$ used for steering on the prompt space remains constant for all the models, regardless of the specification level used during training. Figure 15 demonstrates that the reachability results show almost no difference to those in Figure 6a, indicating that models are robust to variations of the size of $\mathbf{v}_p$ when steering on the prompt space.

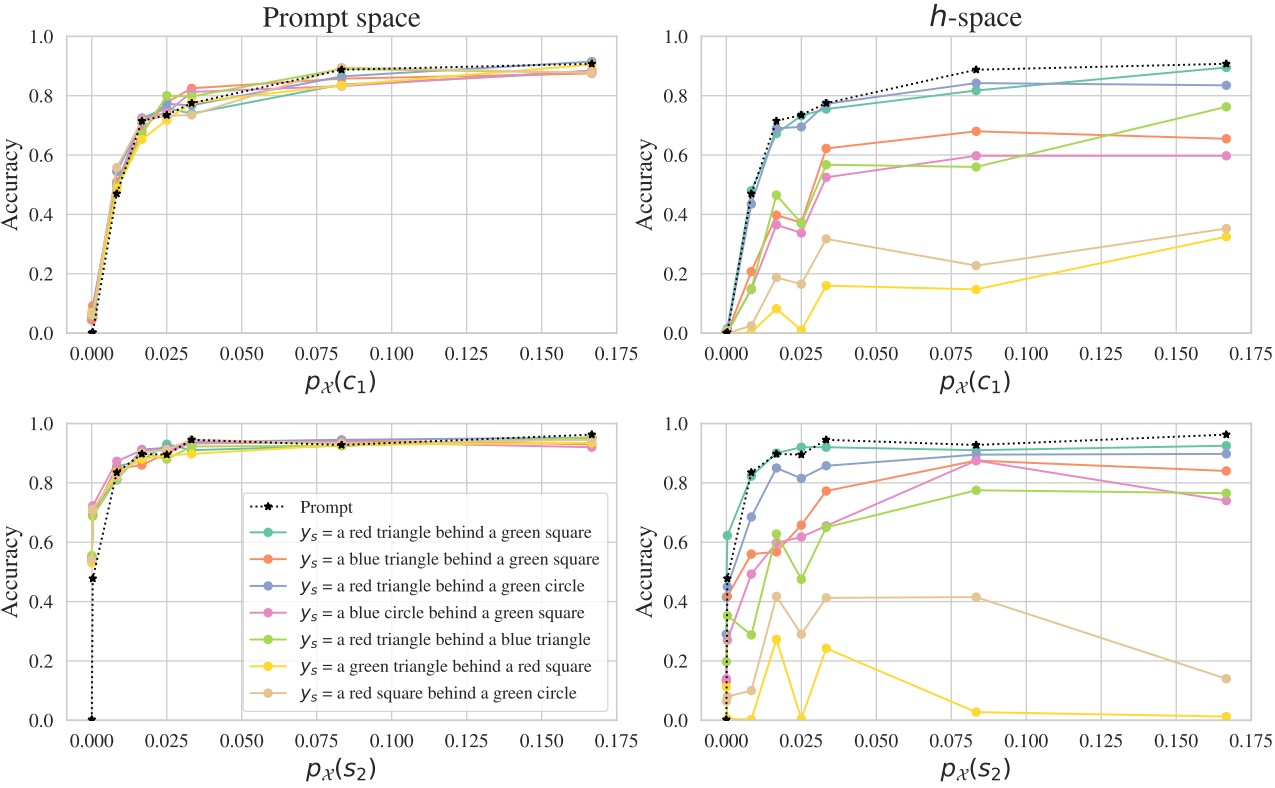

Figure 13: Accuracy of prompting and steering on the prompt space and $h$-space for different starting prompts $y_s$ and varying level of the proportion $p_{\mathcal{X}}$ of images in the train set containing the concepts $c_1 = red$ (top) and $s_2 = square$ (bottom) across the dataset. The target concept combination for all $y_s$ is $(red, triangle, green, square)$.

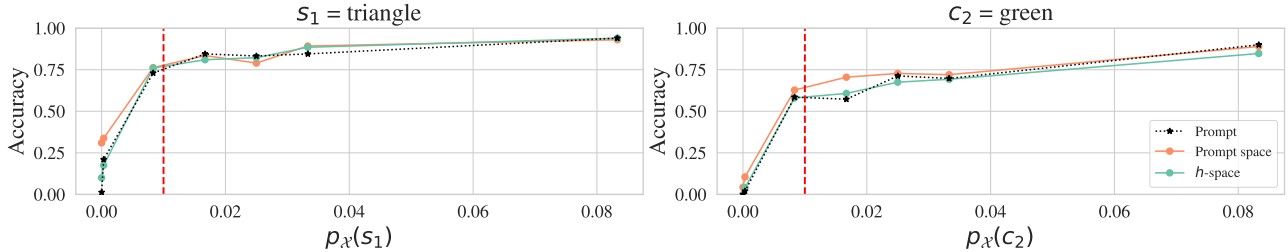

Figure 14: Accuracy of prompting and steering on the prompt space and $h$-space to $(red, triangle, green, square)$ for starting prompt $y_s$ = "a red triangle behind a green square" and varying the proportion $p_{\mathcal{X}}$ of images in the training set containing the concepts $s_1 = triangle$ (left) and $c_2 = green$ (right) respectively. The vertical red line marks the approximate threshold 0.01 that determines the approximate shift in reachability.

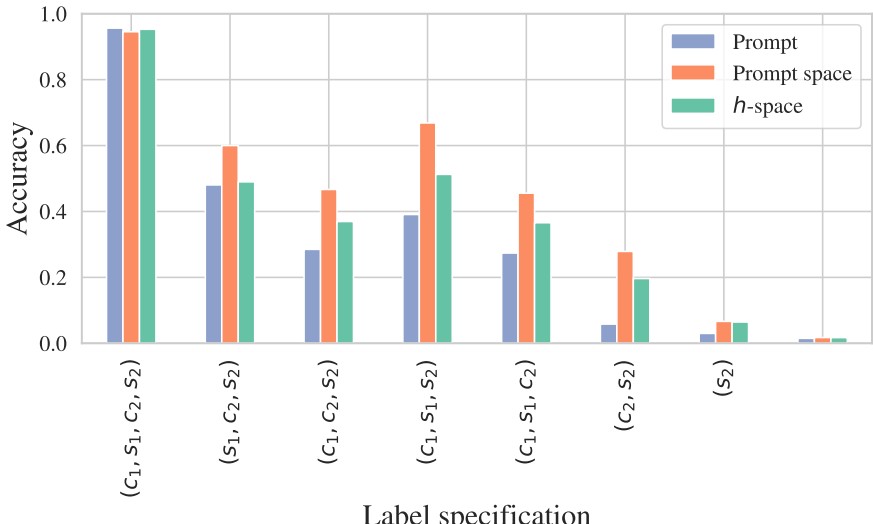

Figure 15: Average reachability across 10 randomly chosen concept combinations $(c_1, s_1, c_2, s_2)$ and varying levels of specification in the captions. Starting prompts $y_s$ used to steer to each of the 10 concept combinations are chosen to fully describe the target concept combinations.

### E.5. Additional Results for Biases

Figure 16 provides an additional example of the scenario studied in Section 5.4. In particular, we tie the concepts $c_2 = red$ and $s_2 = triangle$, and gradually increase the images in the train set containing non-triangular red shapes. Similar to Figure 7, we observe a sharp increase in reachability for red shapes as their representation in the dataset grows.

Additionally, although with more variability than Figure 7, we note a general improvement across all reachability methods on the remaining tied concept (non-red triangles). This suggests that the model learns to disentangle the concept $s_2 = triangle$ as the presence of $c_2 = red$ increases. Moreover, steering consistently outperforms prompting in disentangling the concept $s_2 = triangle$, demonstrating it is a more effective reachability method in such biased scenarios.

## F. Stable Diffusion Implementations

### F.1. Hyperparameters

All experiments on real data are conducted on Stable Diffusion v1.5. We use a DDPM scheduler with $T = 1000$ inference steps to generate images.

Steering is implemented on the prompt space, with $y_s$ describing the target (e.g., "an image of a beaker", "an image of a person walking left"), and a collection $\mathcal{Z}$ of 200 images containing the correct target concepts. The concept vector is initialised at the zero vector and is optimised using Adam for 13 steps. The images required for steering are obtained from openly available datasets such as ImageNet (Deng et al., 2009) and images sampled from Stable Diffusion and DALLE (Ramesh et al., 2022).

### F.2. Train Dataset

Stable Diffusion is primarily trained on subsets of the LAION5B and LAION2B-en datasets (Schuhmann et al., 2022). To estimate the presence of concepts within the dataset, we use unigram and bigram frequencies of images labeled with English captions (LAION2B-en), as provided by Samuel et al. (2024).

The approximate number of captions containing the word *beaker* is 52,000. Despite beakers being easily recognisable and composed of simple visual features, the model struggles to generate them, as shown in Figure 1. We hypothesise that this is due to the relatively low number of images containing beakers compared to the overall dataset size.

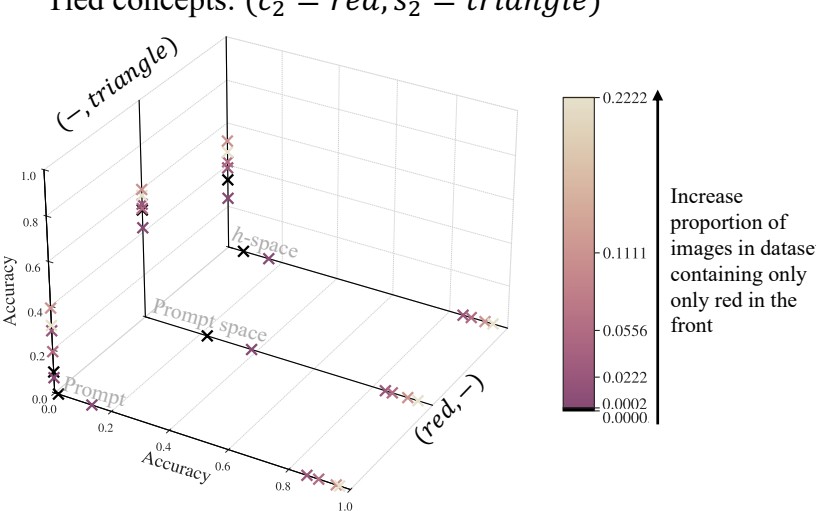

Figure 16: Reachability to concept combinations containing only non-red triangles in the front (vertical axis) and only red non-triangular shapes in the front (horizontal axis), by prompting, steering on the prompt space and steering on the $h$-space. Results are averaged over multiple 6 randomly chosen target concept combinations. When steering, the starting prompt $y_s$ describing those same concepts is used.

The bigrams *walking left* and *walking right* appear in only 431 and 585 captions, respectively, whereas the unigram *walking* appears in over 3,450,000 captions—a significantly higher occurrence. Despite this imbalance, the model can occasionally generate images of people facing either left or right, suggesting an inherent understanding of orientation. However, we hypothesise that the lack of explicit directional specification in most captions limits the model's ability to reliably generate images of people walking in a specific direction.

Finally, the model struggles to generate black-and-white images of rainbows, despite successfully handling black-and-white colour representation in other contexts. We hypothesise that this stems from the scarcity of black-and-white rainbow images in the dataset, making it difficult for the model to disentangle colour information from the broader concept of a rainbow, ultimately revealing a bias in the model's latent space.

### F.3. Steering on the Prompt Space

We compare the images generated through sampling using prompting and steering on the prompt space, ensuring both methods use the same initial random seed. This evaluation is conducted on 50 images per studied concept. Below, we present seven examples illustrating the impact of steering on the final output.

#### F.3.1. SCARCITY OF CONCEPTS

Figure 17 illustrates the effect of steering on image generation towards the target concept: a beaker. We observe that some random seeds initially generate images unrelated to the beaker concept but are successfully guided towards it. In contrast, cases where the model already produces a beaker show minimal modification. Overall, the steered images exhibit attributes commonly associated with beakers, such as transparency, a cylindrical shape, measurement markings, and the presence of liquid.

#### F.3.2. UNDERSPECIFICATION OF CAPTIONS

Figure 18 illustrates the effect of steering towards generating images of a person walking leftward. Our results indicate that steering in this scenario is limited. In many cases, applying steering on the prompt space does not significantly alter the individual's position compared to images generated without steering. While we occasionally observe improvements in the walking direction towards the left, we also find instances where the position shifts towards the right—despite no such

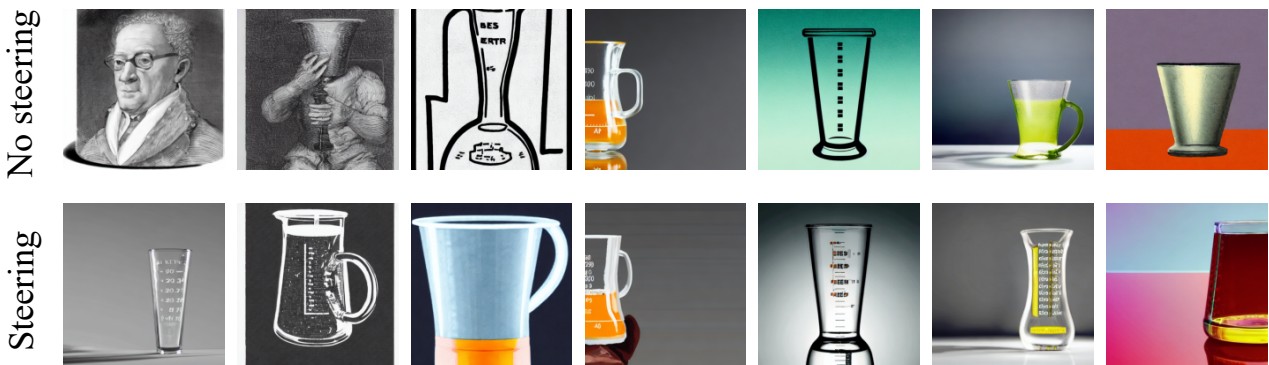

Figure 17: Comparison between sampling on Stable Diffusion with no steering (top row) and with additional steering on the prompt space (bottom row) by using the same random seed. Images are sampled from the prompt "an image of a beaker". The steering vector is optimised between the same starting prompt $y_s =$ "an image of a beaker" and 200 images of beakers.

occurrences in the images used to optimise the steering vector. These findings suggest that reachability is highly constrained when attempting to steer towards concepts that are not explicitly mentioned in the captions. This observation aligns with our discussion in Section 5.3.

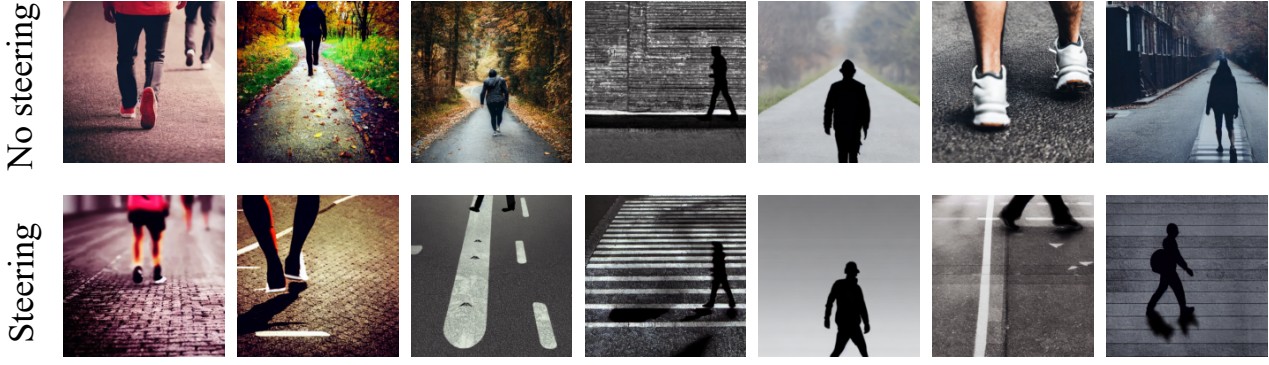

Figure 18: Comparison between sampling on Stable Diffusion with no steering (top row) and with additional steering on the prompt space (bottom row) using the same random seed. Images are sampled from the prompt "an image of a person walking left". The steering vector is optimised between the starting prompt $y_s =$ "an image of a person walking left" and 200 images containing an individual walking towards the left side of the image frame.

### F.3.3. BIASES

Figure 19 illustrates the behaviour of images when steered towards disentangling rainbows from colour. We observe cases where the model reduces the presence of the rainbow, fading the colours from the image, and at times completely removing it. Other images result in an arc-like black-and-white pattern. Interestingly, some steered images resemble black-and-white woodgrain patterns, suggesting a potential bias in the latent space. This may arise because both rainbows and woodgrain share a structure of contrasting light and dark lines, which the model's latent representations might conflate. Finally, in other instances steering proves ineffective, and the model continues to generate images featuring a coloured arc.

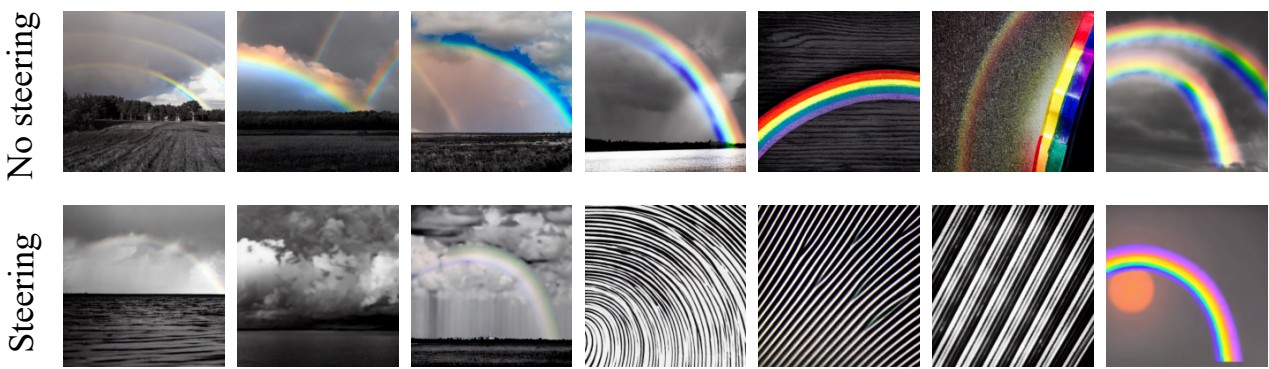

Figure 19: Comparison between sampling on Stable Diffusion with no steering (top row) and with additional steering on the prompt space (bottom row) using the same random seed. Images are sampled from the prompt "an image of a rainbow in black and white". The steering vector is optimised between the same starting prompt $y_s$ = "an image of a rainbow in black and white" and 200 greyscale images containing scenes of rainbows.

## G. CelebA

We conduct a similar analysis to that presented in Section 5 using synthetic data, this time using a subset of 16,000 images of faces from the CelebA dataset (Liu et al., 2015). Leveraging the dataset's existing attribute labels, we assume that the images are generated by underlying factors that take on specific values (concepts), following the structure outlined in Section 3.2. We note that certain factors and biases present in the dataset may remain unaccounted for, as attributes such as posture or lighting are not labelled in the dataset. Furthermore, the positional relations investigated in our earlier experiments that support positional generalisation are not applicable in this context, which may cause deviations from the behaviours observed in synthetic data. Further challenges may also arise from the use of more entangled and context-dependent words in comparison to the simpler captions in the controlled synthetic setup. Nevertheless, this analysis remains insightful for understanding the model's behaviour in more realistic settings.

### G.1. Hyperparameters

We study the attributes gender ($g \in \{\text{man, woman}\}$) and hat ($\hat{h} \in \{\text{wearing a hat, without a hat}\}$). Captions are composed in the form "a $\{g\}$ $\{\hat{h}\}$". For example, "a woman wearing a hat", or "a man without a hat". The balanced dataset is comprised of 4,000 images of each of the four possible concept combinations. The images are transformed using a random horizontal flip ($p = 0.5$) and colour jitter with brightness 0.1, contrast 0.1 and saturation 0.1. As in our previous experiments, when varying the concepts in the train set we approximately preserve the total dataset size. The model architecture is the same as described in Appendix A. We train our models for 400 epochs using the same optimiser and learning rate as before.

The concept vector for steering on the prompt space and $h$-space is obtained using Adam with learning rate 0.02 for 11 steps, and 100 images containing the target concept combinations. To evaluate reachability, we train 4 diffusion models with different random seeds, use them to generate 100 images of the target concept combinations and report the mean results. To help in the evaluations of the generated images, we train two CNNs consisting of three convolutional layers and two linear layers to classify (i) the gender of the person in the image and (ii) if they are wearing a hat or not. Our classifiers obtain an accuracy of 94.6% (gender) and 97.2% (hat) on a held-out validation set consisting of 2,400 images.

### G.2. Baseline

Fixing a starting prompt $y_s$, we apply steering towards the different target concept combinations. For comparison, we also evaluate the reachability achieved through direct prompting of the target combinations. The results are presented in Figure 20a. Additionally, Figure 20b illustrates a comparison of images generated by the model under no steering (prompting $y_s$ = "a woman wearing a hat") and with additional steering towards the concept combination (man, without a hat).

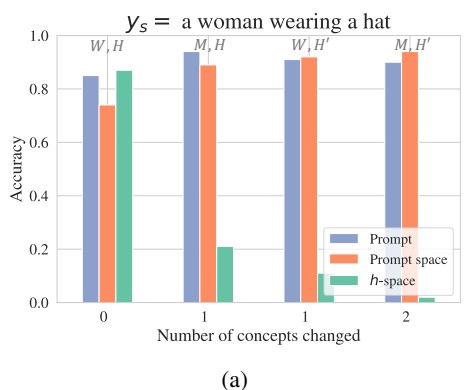

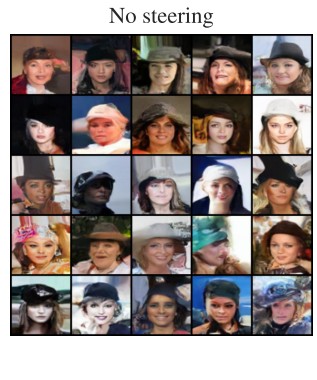

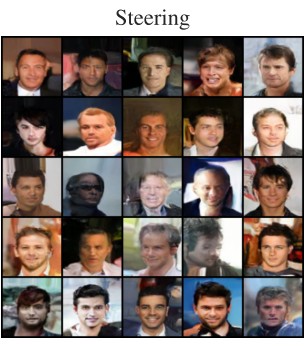

(a)  (b)

Figure 20: a) Reachability to different concept combinations when prompting and when steering from the starting prompt $y_s$ ="a woman wearing a hat". Target combinations are shown at the top of each bar, and are organised according to the number of concepts that differ from the concepts of $y_s$. The notation $M/W$ refers to the gender (man/woman) and $H/H'$ to wearing a hat/without a hat. b) Example of images generated from the prompt "a woman wearing a hat" (no steering) and the images obtained by additionally steering on the prompt space to the concept combination (man, without a hat).

**Concepts remain reachable when steering from diverse starting prompts**  Figure 20a demonstrates that the behaviour observed under balanced conditions aligns with the patterns identified in the synthetic setting. Specifically, while concepts generally remain reachable through prompt-space steering, reachability in the $h$-space declines as the number of altered concepts increases. Overall, steering enables effective access to target concepts, achieving a level of reachability comparable to that of directly prompting the desired concepts. Notably, we observe a systematic decrease in reachability for the concept combination (woman, wearing a hat) relative to the other combinations, which may reflect latent biases in the training data distribution.

### G.3. Scarcity of Concepts

We decrease the presence of the concept $\hat{h}$ = wearing a hat, and consider reachability to the concept combination (man, wearing a hat). Results are presented in Figure 21.

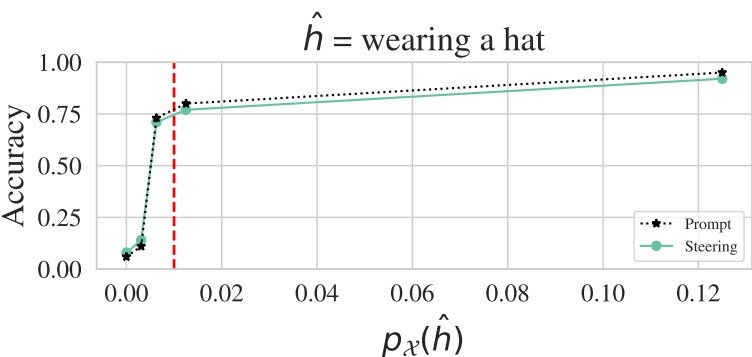

Figure 21: Accuracy of prompting and steering from the starting prompt $y_s$ = "a man wearing a hat" for a varying proportion $p_{\mathcal{X}}$ of images across the dataset containing the concept $\hat{h}$ = wearing a hat. The steering curve shows the results for $\max$ (prompt space, $h$-space). The dotted red line marks the approximate threshold 0.01 of the shift in reachability.

**Reachability drops sharply past a critical threshold**  Figure 21 reveals a threshold pattern consistent with that observed in Figure 5. Specifically, we observe a sharp decline in reachability once the proportion of training images containing a given concept falls below a certain threshold. As in the synthetic setting, the model appears able to learn and generalize a concept even from a small number of examples, provided this minimal threshold is met. This highlights the importance of

ensuring minimal representation of key concepts in training data to ensure reachability.

### G.4. Underspecification of Concepts

We vary the level of specification of captions of the training dataset, and in particular reduce the captions of the form "a $\{g\}$ $\{\hat{h}\}$" to the following:

1. Remove $g$: replace the caption with "a person $\{\hat{h}\}$"

2. Remove $\hat{h}$: replace the caption with "a $\{g\}$"

3. Remove $g$ and $\hat{h}$: replace the caption with the empty string, ""

We steer from the starting prompt describing only seen concepts. For example, when removing $\hat{h}$ from the captions, to steer to (woman, wearing a hat) we use $y_s = $ "a woman". Additionally, we compare the accuracy of prompting using the complete description of the target concepts. Results are presented in Figure 22a.

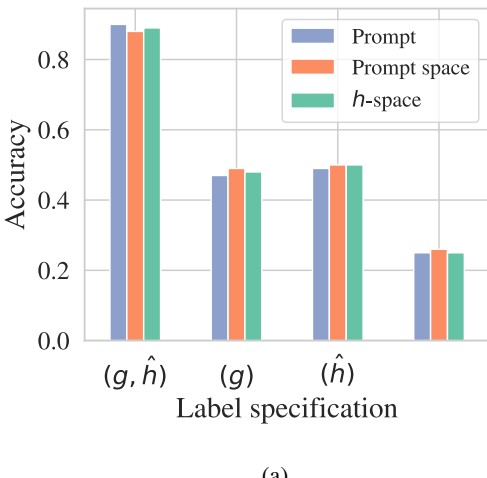
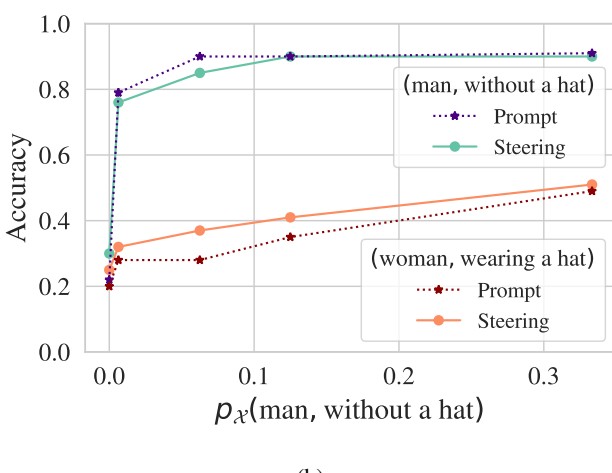

(a)

(b)

Figure 22: a) Average accuracy of prompting and steering to the different concept combinations for different levels of concept specification. b) Accuracy of prompting and steering to the concept combinations (man, without a hat) and (woman, wearing a hat) under biased conditions. Starting prompts $y_s$ describe the target concept combinations, while varying the presence of the concept combination (man, without a hat). The steering curve shows the results for $\max$ (prompt space, $h$-space).

**Underspecification hinders reachability**   Similar to Section 5.3, we observe a rapid decrease in the reachability of concepts. The reachability is observed to be close to the value of sampling the unspecified concepts randomly: when one factor is unspecified, reachability is approximately 50%, and when both are unspecified, reachability is approximately 25%.

### G.5. Biases

We tie the concepts "man" and "wearing a hat" (which also causes the concepts "woman" and "without a hat" to be tied), and gradually increase the presence of images containing the concept combination (man, without a hat), thus reducing the bias. We evaluate reachability to (man, without a hat) and (woman, wearing a hat). Results are presented in Figure 22b.

**Increasing the presence of an individual concept increases separate reachability to both concepts**   Figure 22b shows a clear increase in reachability to either concept combination as the bias in the dataset is decreased. Most noticeably, we observe a sudden increase in reachability to (man, without a hat), similar to the threshold patterns previously observed. Moreover, the reachability to (woman, wearing a hat) also gradually increases, suggesting that the models become more capable of disentangling concepts. We further note that this concept combination is consistently more reachable through steering.

