# OpenReview forum: "Concept Reachability in Diffusion Models: Beyond Dataset Constraints"
_ICML.cc/2025/Conference — ICML 2025 poster_

### Official Review · Reviewer_saQX · 2025-03-11

**Overall Recommendation:** 4

**Summary:**

This paper compares the effects of prompting vs. steering on diffusion models, and in particular how well these approaches allow for the control of specific concepts in the generated output.  To provide control in the experiments, a synthetic dataset of overlapping shapes is created, and the concepts of interest are the types of shape in the image and the colors of the shapes.  The paper then explores the effects of bias (frequent co-occurance of a shape/color pair), concept scarcity (the degree to which the concept has been observed), and caption specification (how well the caption associated with an images fully captures the concepts in the image).  The main finding is that commonly used prompting is not sufficient to realize concepts in the presence of badly formed data, whereas steering allows better generation of scarce concepts and allows the disentanglement where spurious correlation/bias exists.  Furthermore, prompt steering rather than steering in the latent space of the model is shown to be more robust and reliable.

### Update
Following the rebuttal I have maintained my score and recommend accepting the paper.

**Claims And Evidence:**

Yes.  The use of the synthetic data allows specific control over the presence of concepts, and degree to which the prompt specifies the concepts.  The are examples of showing the equivalent effects on real-world samples, albeit limited in number.

**Essential References Not Discussed:**

I have no concerns here.  The assume the authors make is that if a concept cannot be reached by steering vectors, then it is not reachable.  However, there are other methods for steering the outputs generated by models, and these might perhaps be able to tease out the concept from the model.  I do think it is fine to focus only on steering vectors, but this limitation/assumption might be worth spelling out.

**Experimental Designs Or Analyses:**

I have no concerns about the experiments, but had a minor question regarding using the accuracy of a classifier to measure concept reachability.  Does the underlying accuracy of the classifier not need to serve as the baseline for the reachability measure?  For example, if the classifier reports 90% accuracy on the generated data, reachability is reported at this level — however if the classifier itself is only 90% accurate, then might the concept not be considered full reachable?

**Methods And Evaluation Criteria:**

Yes.  The use of the synthetic data removes ambiguity and noise present in real-world samples (such as the degree to which a concept is present in an image, or how other the presence of concepts affect interpretation of a concept).

**Other Comments Or Suggestions:**

In Section 4.3:  replace “with not shapes” with “with no shapes”

In Section 4.3:  replace “report the average results” with “report the mean results”

**Other Strengths And Weaknesses:**

Strengths
+ The authors will release the code upon accept to aid reproducibility.
+ The experiments are well put together and results support the claims made.

Weaknesses:
- Perhaps more analysis on trickier real-world cases would make the paper stronger.

**Questions For Authors:**

Q1:  Does the degree to which the shapes overlap affect the reachability measure?  For example, a heavily occluded background shape might be harder for the classifier to identify, even if the shape itself is correct?  This could mean that reachability is under-estimated in more difficult cases, as opposed to the concept being less reachable.

Q2:  In Figure 4 — are the shape/color combinations at the top of figure only representative examples, or does the figure relate to these exact samples?

**Relation To Broader Scientific Literature:**

The paper references related work that these experiments build on, and relevant citations are included to support statements made.

**Theoretical Claims:**

N/A.  The paper presents an empirical study.

---

> ### Author Rebuttal · Authors · 2025-04-01
>
> Thank you for taking the time to review our work. We are very happy to read your positive review! We have read through the comments and suggestions you made and addressed them below:
> ___
> - **Does the underlying accuracy of the classifier not need to serve as the baseline for the reachability measure?**
>
> When evaluating on human-labelled images (5400) that were generated by prompting from two diffusion models trained on balanced data, the classifier obtains an accuracy of 96.63%, hence the results would have to consider the possible error of the classifier. We consider this accuracy to be sufficient to reflect the impact of our experiments on reachability. We will add a clarification about this to the paper.
> - **Paper focuses on steering, but there are other methods for reaching the concept than steering**
>
> This is fully correct: steering falls under the broader category of modifying the latent space and many variations exist on how this latent space modification is performed (which model features are being modified, how this modification is performed). Our focus here was on steering as it has achieved successful results in both diffusion models and autoregressive modelling, while still being relatively simple and efficient to implement (and hence promising also for practical use-cases).
> - **More complex data**
>
> This is addressed in the response to qawW (CelebA).
> - **Typos in Section 3.4**
>
> Thank you for pointing these out! We will correct this in the updated version of the paper.
> - **Does the degree to which the shapes overlap affect the reachability measure?**
>
> Originally, we encountered challenges when classifying the back shape in case these were heavily occluded. We modified the neighbourhood within which the front shape was sampled in order to ensure a reasonable portion was visible. The minimum percentage of each back shape visible in the balanced train set for the diffusion models is the following: circle 52.46%, triangle 48.97%, square 59.38% - this will be added to the Appendix in the dataset details. Overall, approximately half or more, of each shape is visible. We did not explore this further, however after this modification we didn’t notice any significant difference in reachability with the different shapes in the back.
> - **In Figure 4 — are the shape/color combinations at the top of figure only representative examples, or does the figure relate to these exact samples?**
>
> These are the colour-shape combinations for the accuracies displayed in the graph. For example, the first column shows a green triangle behind a red triangle, which changes no concepts with respect to the starting prompt y_s. Note that it does not refer to the exact relative position between the two shapes, only the combination (we will clarify this).

---

> > ### Comment · Reviewer_saQX · 2025-04-03
> >
> > Thank you for the follow up. I found the paper interesting and look forward to seeing the results on the CelebA data. I will maintain my recommendation to accept the paper.

---

### Official Review · Reviewer_fxhd · 2025-03-13

**Overall Recommendation:** 3

**Summary:**

This paper studies the concept reachability in diffusion models and focuses on the effects of three common constraints in datasets: scarcity of concepts, underspecification of captions, and biases. The work shows that although some concepts are reachable for the model, prompting fails to provide sufficient information to reach them. In addition, this paper proposes steering as a novel controlling mechanism for better concept reachability in the generation of diffusion models.

## update after rebuttal
I would like to keep my original rating. This paper presents insightful findings, but my main concerns regarding evaluation on more complex datasets and clarification on general image generation remain insufficiently addressed during the rebuttal. The authors acknowledged the usage of CelebA in the response to Reviewer qawW but failed to provide results during the rebuttal.

**Claims And Evidence:**

The claims are successfully supported by experiments on controlled synthetic data.

**Essential References Not Discussed:**

I didn't identify any such references.

**Experimental Designs Or Analyses:**

The experiment design is convincing overall, except for several concerns: (1) The experiments focus on the analysis of synthetic data and only show three examples for three challenges for the application in image generation with diffusion models. Is it possible to evaluate the performance on a larger scale for diffusion model generation? (2) How is the efficiency of steering in terms of application?

**Methods And Evaluation Criteria:**

The proposed methods and evaluation criteria make sense under the problem setting.

**Other Comments Or Suggestions:**

The notion of Figure 7 is unclear to me, and I didn't fully understand how Figure 7 supports the claims regarding biases.

**Other Strengths And Weaknesses:**

Strengths:

This paper is well-written and the contribution is valid by categorizing and analyzing the common constraints in concept reachability. The steering approach is effective in these settings and image generation of diffusion models. The experiments on the synthetic data are comprehensive, with settings that are well-controlled for each constraint.

Weaknesses:

This work mainly focuses on the analysis of synthetic data, lacking broader evaluation for more general image generation with diffusion models.

**Questions For Authors:**

Please see the questions in Experimental Designs Or Analyses. Additional question: Will the synthetic dataset be too simplified? CLEVR [1] is a dataset with multiple combinations of attributes with more realistic appearances, and the code to render new images is provided as well. Will CLEVR be a better choice for analysis?

[1] Johnson, Justin, et al. "Clevr: A diagnostic dataset for compositional language and elementary visual reasoning." Proceedings of the IEEE conference on computer vision and pattern recognition. 2017.

**Relation To Broader Scientific Literature:**

Previous work has proposed to optimize the input of specific task, or introduce steering vectors at specific layers to improve the fine-grained control in LLM or text-to-image model generation. This work studies the concept reachability of steering the textual prompt or in the U-Net bottleneck layer under three scenarios of dataset constrains.

**Theoretical Claims:**

There is no major theoretical claim in this paper.

---

> ### Author Rebuttal · Authors · 2025-04-01
>
> Thank you for reading through our paper! We really appreciate your positive feedback, and have addressed the questions/comments you made below:
> ___
> - **How is the efficiency of steering in terms of application?**
>
> Our current work did not focus on analysing the computational efficiency of steering precisely, but overall we found that the optimisation of the steering vectors is dependent on the dataset and learning rate used for optimising, with the number of steps required to implement the optimisation varying. Analysing the efficiency of steering under different conditions in large-scale models would be an interesting extension. In Stable Diffusion, the optimisation of the steering vector required less than 20 steps.
> - **CLEVR and synthetic data**
>
> Although the Clevr dataset contains more complex images with greater detail and 3D structure, we believe that repeating the experiments with Clevr would yield similar results to those already observed within our synthetic framework. The underlying factor structure of both datasets is essentially the same, and as such, we anticipate that the observed outcomes would not vary significantly. In our work, we opted for a simpler dataset to ensure controllability, robustness, and realisable methods for evaluating accuracy, as well as more scalable model training at a lower computational cost. We further address experiments with more complex data in the response to Reviewer qawW (CelebA).
> - **Figure 7**
>
> Figure 7 shows the reachability to images containing either only circles or only blue shapes in the back, under conditions where these two concepts are tied. As the presence of images containing only blue in the back is increased (thus, reducing the bias), the reachability of images containing only blue in the back increases - this is shown in how the light-coloured X's produce a higher accuracy on the horizontal plane. However, this also leads to a general trend (with some variability) of increase in the accuracy of images containing only circles in the back - note that generally across all methods, on the vertical axis, the lightest coloured X's achieve the highest accuracy. We will clarify the description of this Figure in the paper.

---

> > ### Comment · Reviewer_fxhd · 2025-04-03
> >
> > Thanks for the rebuttal from the authors. I would like to keep my original rating with the following concerns. This paper presents interesting findings and I believe this paper can benefit from experiments with more complex data. As reviewer qawW mentioned, including results with CelebA could make this work more convincing. In addition, the rebuttal did not include my concerns about the more general image generation. Clarifying the scope of this work and providing more details on practical usage (such as Fig 8) could be helpful for future work.

---

### Official Review · Reviewer_qawW · 2025-03-13

**Overall Recommendation:** 3

**Summary:**

This paper explores the influence of three core dataset issues on concept reachability in text-to-image diffusion models. Through a synthetic setup, the paper constructs dataset variations corresponding to the three dataset issues and test concept reachability by evaluating the concepts in generated images after training on the dataset variation. The paper introduces novel perspectives into how data issues impact diffusion model generation and offers insights into the benefit of steering in image generation controls.

**Claims And Evidence:**

The claims made are clear and supported with one small issue: maybe I missed this in the paper, but the evidence behind the claim that overly detailed prompts may not help with reachability on line 373 is not available.

**Essential References Not Discussed:**

It can be interesting and relevant to discuss the connection between (1) the diminishing reachability as the dataset issues worsen and (2) data attribution for the diffusion models trained on the problematic datasets, such as [1].

[1] Wang, Sheng-Yu, et al. "Evaluating data attribution for text-to-image models." Proceedings of the IEEE/CVF International Conference on Computer Vision. 2023.

**Experimental Designs Or Analyses:**

1. The choice of starting prompt y_s seems rather arbitrary. What is the rationale behind the selected y_s for each experiment (e.g., how much does it deviate from y_e and why)?
2. The quality of concept classifiers is not shown in the paper. How well does it serve as the evaluator for reachability accuracy?

**Methods And Evaluation Criteria:**

The method works effectively for the proposed synthetic setup. Although Sec 5.5 and Appx F.3 show the success of steering on Stable Diffusion, these are essentially verifying concept customization but not how dataset issues impact the reachability in general in real world distributions. However, the effectiveness in real-world dataset can be readily tested, e.g., CelebA offers labels of attributes for subsampling to construct the dataset issues.

**Other Comments Or Suggestions:**

Figure 11 in Appx E is missing a legend for dots in the plot, which makes it a bit confusing to read the figure.

**Other Strengths And Weaknesses:**

The paper is clearly presented with substantial visualizations and graphs. The proposal enables novel perspectives to root cause reachability in the dataset.

**Questions For Authors:**

1. Despite that Sec 5.5 and Appx F.3 show the success of steering on Stable Diffusion, these are essentially verifying concept customization but not how dataset issues impact the reachability in general in real world distributions. However, the effectiveness in real-world dataset can be readily tested, e.g., CelebA offers labels of attributes for subsampling to construct the dataset issues. The utility of the discovery in this paper can be more impactful if shown on realistic settings.
2. The choice of starting prompt y_s seems rather arbitrary. What is the rationale behind the selected y_s for each experiment (e.g., how much does it deviate from y_e and why)?
3. The quality of concept classifiers is not shown in the paper. How well does it serve as the evaluator for reachability accuracy in the synthetic setting?
4. The image illustrations in Figure 3 and Figure 4 present different positions of shapes and portions of overlapping. Does the IoU, especially the visible portion of the back shape, have any impact on the result?
5. It is interesting to see in Figure 6(b) that prompt space and h-space optimization lead to different performance on red and blue colors. Is there any hypothesis why this happens?

**Relation To Broader Scientific Literature:**

This paper is a novel addition to the data-centric reachability study in diffusion models and offers insights and recommendations for future improvement of text-to-image generative models and generation controls.

**Theoretical Claims:**

No proof or theoretical claims

---

> ### Author Rebuttal · Authors · 2025-04-01
>
> Thank you for your thorough review and thoughtful comments! We address your questions below:
> ___
> - **Evidence behind the claim that overly detailed prompts may not help with reachability**
>
> The starting prompt y_s used to implement the steering is as described in Section 3.5, however the accuracy displayed in Figure 6(a) for prompting is that of the prompt containing the full description of the target concepts. Hence, the graph shows that as the specification of concepts is decreased, the reachability of prompting the full description (which is overspecified with respect to the seen specification level) is severely affected due to the specification of the train set being limited. This will be clarified in the paper.
> - **Rationale behind the selected y_s for each experiment (e.g., how much does it deviate from y_e and why)?**
>
> In Experiment 1 we vary the deviation (number of concepts changed) between y_s and y_e to understand the behaviour between lower to higher changes in number of concepts. In the remaining sections we choose the case y_s = y_e, as it overall showed the most stable performance with respect to steering on the h-space (prompt space is mostly unaffected), and would be the natural choice when trying to reach a concept in a real data model. That is, if generating something on a model via prompting fails, in most cases one would to try to additionally steer on the prompt that describes the desired outcome.
> - **Quality of the classifier**
>
> This is addressed in the response to Reviewer saQX.
> - **Reference suggestion**
>
> Thank you for the reference! Data attribution methods are certainly related to our setup. We will discuss data attribution methods in diffusion models and the reference provided in the updated version of the paper.
> - **Figure 11**
>
> We will add a legend labelling the random seed to clarify the figure.
> - **Portion of visible back shape**
>
> This is addressed in the response to Reviewer saQX (Degree of overlap). Note that the illustrations in Figure 3 and Figure 4 are diagrams - actual samples of the train set are provided in Figure 10 in the Appendix.
> - **Difference between h-space and prompt space in Figure 6(b)**
>
> Figure 6(b) shows an example for one model of the behaviour of prompting and steering when trying to reach red in the back when the label c_1 is not specified in training, or reaching a square in the front when the label s_2 is not specified. We wanted to note that, particularly in the case of prompting and steering on the h-space, the generated output is close to randomly sampling the value of the unspecified label (in the case of c_1, the colour of the front shape in the target combination is green and so there are only two choices for the back colour). Steering on the prompt space, instead, produces higher accuracy on the target combination, leading to a higher value achieved on the top axis (red and square, respectively) than the one achieved by prompting, although remaining close to this value. We will update the explanation of this in the paper. The dimensionality, level of disentanglement, and the dependency of the h-space on the timestep $t$ could potentially impact the observed differences.
> - **CelebA**
>
> We agree that extensions to real-world data will be valuable. We are currently implementing our framework on the CelebA dataset to assess whether its structure aligns with the properties required for our analysis. As part of this, we are constructing approximately 15-20 dataset variations and training multiple models per dataset. Given the compute we have available we hope to be able to present the full results before the end of the author-reviewer discussion period and include them in the paper.
>
> We currently have observed similar trends in a balanced dataset as in Experiment 1: reachability on the prompt space is more consistent than reachability on the h-space, which is affected by number of factors changed. Moreover, we have also observed a decrease in reachability as the level of specification of the train set is decreased. We plan to include this in the paper, as well as an analysis of scarcity and biases, with detailed examples of steered images.
>
> We remark that while CelebA does provide attribute labels for selected factors, other latent factors such as background, lighting and pose are uncontrolled, and it is only the use of synthetic data that guarantees a fine-grained cause effect study of the impact of dataset modifications.

---

> > ### Comment · Reviewer_qawW · 2025-04-04
> >
> > Thank you for your detailed explanation. I appreciate the clarifications provided and, after careful consideration, I will keep my recommendation of weak accept.

---

> > > ### Author Response · Authors · 2025-04-07
> > >
> > > Thank you to all reviewers for their positive recommendations! We would like to share the obtained results on CelebA. By looking at two characteristics (male/female and hat/no hat), we defined a gradual modification of the dataset. We train 4 diffusion models on each train set, and evaluate accuracy using two classifiers. Below we summarise the main conclusions obtained on synthetic data and their generalisation to CelebA:
> > >
> > > - **Concepts remain reachable when steering from diverse starting prompts**
> > >
> > > We used a fully balanced dataset where each of the studied combinations was equally seen during training. We fix one starting prompt and vary the reachability to different concept combinations. We observe the different concept combinations to be reachable via steering, and in particular, steering on the h-space is observed to depend on the number of concepts changed, similarly to Figure 4 in synthetic data.
> > > - **Reachability drops sharply as concepts become more scarce**
> > >
> > > We decrease the number of images containing people wearing hats, and target the generation of men wearing hats. Similar to the observation on synthetic data (Figure 5), we observe a critical threshold (in terms of the number of images containing the hat concept) below which reachability significantly drops. We also identify that it is possible to reach the concept by steering in settings where prompting does not work effectively.
> > > - **A decrease in underspecification hinders reachability**
> > >
> > > A decrease in specification of captions significantly decreases reachability across all methods. We compare reachability when specifying both concepts, one of the concepts, or neither. When specification of one concept is removed, accuracy is approximately 50%, when specification of both captions is removed, accuracy is close to 25%.
> > > - **Increasing the presence of an individual concept increases separate reachability to both concepts**
> > >
> > > We tie the concepts of female and not wearing a hat, and target the generation of females wearing hats and males not wearing hats, thus aiming to break the bias. As we gradually increase the presence of males not wearing hats in the dataset, we observe a rapid increase in reachability to males not wearing hats (expected), but also a general increase in reachability to females wearing hats across all reachability methods. This concept combination is, in particular, most reachable through steering.
> > >
> > > We will add a section in the Appendix to present these results.

---

### Official Review · Reviewer_7wCq · 2025-03-16

**Overall Recommendation:** 3

**Summary:**

This paper focuses on the limitations of prompting for model control. It shows how steering is a more robust mechanism to enhance concept reachability. The authors study three common dataset issues: concept scarcity, underspecification of captions, and biased co-occurrence of concepts. Experiments are evaluated mostly on a controlled synthetic dataset (colored shapes with specific positional relationships). Their findings demonstrate that steering vectors significantly improve concept reachability, particularly in out-of-distribution or underspecified conditions, and they support this with empirical studies including extensions to real data using Stable Diffusion.

**Claims And Evidence:**

The main claims are supported by well-designed experiments.

**Essential References Not Discussed:**

Essential references are discussed.

**Experimental Designs Or Analyses:**

The experimental design is clean and systematic.

**Methods And Evaluation Criteria:**

The methodology is appropriate for analyzing robustness of model control. The synthetic dataset allows for precise control over concepts, facilitating clear analysis of reachability mechanics.

**Other Comments Or Suggestions:**

- If possible, figure 9 should be moved to the main paper.
- The notation for the steering vectors $\mathbf{c}_p$ and $\mathbf{c}_h$ can be confused with the concept notation $c$. The authors should use for example $\mathbf{s}_p$ and $\mathbf{s}_h$.

**Other Strengths And Weaknesses:**

### Strengths

- The empirical setup is clean and contributes to the steering literature.

- Insightful findings for alternative to prompting for model control.

- Relevance to practitioners seeking more controllable diffusion model behavior.

- The papers is really well-written and organized.

### Weaknesses

- While results are robust within their synthetic setup, the paper would benefit from further discussion on the generalization of these findings to real-world, more complex datasets.

- See questions below.

**Questions For Authors:**

- Are the notations $[f_{i_1}, f_{i_2}, \ldots, f_{i_j}]_X$ and $[f_1^{(i)}, f_2^{(j)}, f_3^{(k)}]_X$ the same?

- Can you better explain the difference between $y_e$ and $y_s$? Which one is the target combination? And Why do you need $\mathbf{x}_0$?

- Figure 4, why steering from the initial prompt "a green triangle behind a red triangle" to a completely different combination such as "a red circle behind a blue square"? Isn't it a too "extreme" steering?

- Can you provide more examples on real datasets?

**Relation To Broader Scientific Literature:**

The connection to steering literature (including LLMs and diffusion models) is well established.

**Theoretical Claims:**

The paper is primarily empirical. There are no formal theorems or proofs.

---

> ### Author Rebuttal · Authors · 2025-04-01
>
> Thank you for reading through our work and providing your feedback! We have read through your comments and address your questions below:
> ___
> - **Notation of steering vectors**
>
> We noticed that the notation $\mathbf{s}$ may also lead to confusion due to the labels $s_1$ and $s_2$ for the factors in our dataset. However, we can change the notation for the steering vectors to $\mathbf{v}$.
> - **Notation for $[f_{i_1}, f_{i_2}, \dotsc, f_{i_k}]_X $ and $[f_1^{(i)}, f_2^{(j)}, f_3^{(k)}]_X$**
>
> The notation **$[f_1^{(i)}, f_2^{(j)}, f_3^{(k)}]_X$** refers specifically to the diagram. This diagram contains only three factors, so the notation refers to the combination of images that contain one specific value or concept for each factor in the dataset. The notation $[f_{i_1}, f_{i_2}, \dotsc, f_{i_j}]_X$ refers to the $n$-factor case, and only fixes the concept value for a subset of the factors, indexed as $i_1, i_2, \dotsc i_j$. We will modify the indexing in the diagram to avoid confusion.
> - **Difference between $y_e$ and $y_s$**
>
> $y_e$ describes the properties of the images we steer to (the end target). $y_s$ is the prompt from which we start, that may (if $y_s = y_e$) or may not describe the desired properties. $\mathbf{x}_0$ is the collection of images containing the target concepts that are used to steer the generation process (they may have been generated by sampling from a model using the prompt $y_e$ or obtained from a test set).
> - **Figure 4: why steer to such extreme combinations?**
>
> We wanted to explore the behaviour in extreme modifications under balanced conditions in order to understand the different reachability methods. Perhaps it would be expected that as the number of factors changed (with respect to $y_s$) increases, reachability via steering decreased. However, this is not the case in steering on the prompt space, which highlights structural differences in the spaces where steering is implemented. Throughout the remaining experiments presented in the main body, the steering is implemented in the case $y_s = y_e$, which we found to produce the most stable results for steering on the $h$-space. We will clarify this in the revised version of the paper.
> - **Examples on real datasets**
>
> This is addressed in the response to Reviewer qawW (CelebA).

---

### Decision · Program_Chairs · 2025-05-01

**Decision:**

Accept (poster)

**Comment:**

This work proposes a thorough analysis on the topic of concept reachability, focusing on text-to-image diffusion models. For that, they design a synthetic (fully controllable) dataset that allows training under the following situations: scarcity of concepts, underspecification of captions, and biases.
The results show that steering is more effective at reaching concepts under such underrepresented situations, allowing better disentanglement of concepts when bias is present (suprious correlations).


The rebuttal has been engaging and I thank both authors and reviewers for that. Many questions have been solved. The reviewers agree in that this work is very clearly presented, with a thorough evaluation.  However, reviewers fxhd, saQX, qawW, 7wCq  found that more evaluation on realistic data (eg. CelebA) would be beneficial. I also agree with this observation. The authors mention they are running experiments on CelebA, and provide initial results. They commit to add CelebA in the final paper.


Given the actual scores and commitment of the authors to include CelebA, I believe this paper would reach the quality required for ICML. I think that the scope and findings of this paper will be valuable for the community, given the strong use of T2I Diffusion and the need to satisfy user needs beyoned the nominal behavior (reach any concept).